# LARGE CONVOLUTIONAL MODEL TUNING VIA FILTER SUBSPACE

**Wei Chen, Zichen Miao, Qiang Qiu**
Purdue University, IN, USA
{chen2732, miaoz, qqiu}@purdue.edu

## ABSTRACT

Efficient fine-tuning methods are critical to address the high computational and parameter complexity while adapting large pre-trained models to downstream tasks. Our study is inspired by prior research that represents each convolution filter as a linear combination of a small set of filter subspace elements, referred to as *filter atoms*. In this paper, we propose to fine-tune pre-trained models by adjusting only filter atoms, which are responsible for spatial-only convolution, while preserving spatially-invariant channel combination knowledge in atom coefficients. In this way, we bring a new filter subspace view for model tuning. Furthermore, each filter atom can be recursively decomposed as a combination of another set of atoms, which naturally expands the number of tunable parameters in the filter subspace. By only adapting filter atoms constructed by a small number of parameters, while maintaining the rest of model parameters constant, the proposed approach is highly parameter-efficient. It effectively preserves the capabilities of pre-trained models and prevents overfitting to downstream tasks. Extensive experiments show that such a simple scheme surpasses previous tuning baselines for both discriminate and generative tasks[1].

## 1 INTRODUCTION

Large models have demonstrated exceptional performance across diverse domains and tasks (Brown et al., 2020; Dosovitskiy et al., 2020; He et al., 2016; Kirillov et al., 2023; Radford et al.; Rombach et al., 2022; Silver et al., 2016; Touvron et al., 2023; Vaswani et al., 2017), attributing to their capability to effectively represent complex patterns and relationships (Khan et al., 2020) by pre-training on massive datasets (Russakovsky et al., 2015; Raffel et al., 2020; Zhu et al., 2015). A common strategy to adapt these large models for specific downstream tasks is fine-tuning them with full parameters. But this method presents two main challenges: (1) Adjusting a vast number of parameters for particular target tasks is computationally intensive; (2) The limited availability of target data increases the risk of overfitting (Lian et al., 2022).

To address these challenges, researchers have developed parameter-efficient methods (Chen et al., 2022; Hu et al., 2021; Jia et al., 2022; Shen et al., 2021; YEH et al., 2023; Zaken et al., 2022) by fine-tuning the pre-trained models with only a minimal number of parameters. Among these methods, LoRA (Hu et al., 2021) fine-tunes models without altering the model architecture, becoming notably popular for its efficacy. However, LoRA still risks overfitting when fine-tuned on limited data and compromising the generalization capability of large models. For instance, Figure 1 illustrates that with only 5 training samples, LoRA tends to produce images that closely resemble the training data, compromising the ability for diverse image generation, compared with pre-trained models.

**Motivation.** To preserve the capabilities of pre-trained models when fine-tuning them on the downstream tasks, one prominent approach in continual learning (Parisi et al., 2019; Rusu et al., 2016; Yoon et al., 2019) is to formulate convolution filters in ConvNets as a linear combination of filter atoms (Li et al., 2019; Papyan et al., 2017; Qiu et al., 2018a) and fine-tuning only filter atoms (Miao et al., 2021a; Zhai et al., 2021). Specifically, filters in each convolutional layer are decomposed over a small set of filter subspace elements, referred to as *filter atoms*, responsible for

---

[1]Our code is available at: https://github.com/weichennone/convnet_finetune

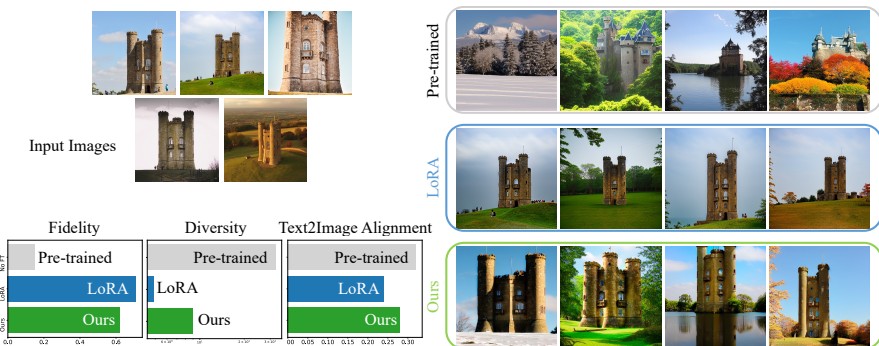

Figure 1: Compared with LoRA (Hu et al., 2021), our method updates a small set of parameters and mitigates the risk of overfitting to the target concept. In this example task of learning the concept ⟨castle⟩ from 5 input images, we only require fine-tuning **0.75M** parameters, a stark reduction from the 22.67M parameters required by LoRA. Furthermore, the model fine-tuned by our method captures the target concept while ensuring a rich diversity and strong alignment with input text prompts. It demonstrates that our approach preserves the generalization capability of large models. The text prompts used to generate images from left to right are: "The ⟨castle⟩ stands against a backdrop of snow-capped mountains", "A ⟨castle⟩ surrounded by a lush, vibrant forest", "The ⟨castle⟩ overlooks a serene lake", and "The ⟨castle⟩ in the autumn season with colorful foliage".

*spatial-only convolution.* Each convolutional layer is now constructed as linear combinations of filter atoms using decomposition coefficients, referred to as *atom coefficients*, responsible for the *spatially invariant channel combination*. Hypothesizing variations across tasks can be reduced by bridging spatial discrepancies in the images, we propose to calibrate the pre-trained model by solely fine-tuning the spatial-only filter atoms while preserving the spatially-invariant channel weights, *i.e.*, atom coefficients.

In our work, we demonstrate that fine-tuning a large model via filter atoms is substantially effective and parameter-efficient, as filter atoms are responsible for spatial-only convolution and usually comprise only a few hundred parameters. This strategy is in harmony with task subspace modeling principles, which suggest that task parameters occupy a low-dimensional subspace, allowing tasks to be represented as combinations of latent basis tasks (Evgeniou & Pontil, 2007; Kumar & Daume III, 2012; Maurer et al., 2013; Romera-Paredes et al., 2013; Zhang & Yang, 2021). We also discover that maintaining fixed atom coefficients, *i.e.*, spatially-invariant channel mixing weights, plays a crucial role in preserving the generalization capability of pre-trained large models.

With a large number of parameters fixed, fine-tuning only a tiny set of parameters in filter atoms is potentially challenging to adapt to more complex tasks. We further demonstrate a simple yet effective way to expand the tunable parameters in filter subspace, without any modification on atom coefficients, by decomposing each filter atom over another set of filter atoms. This process provides an *overcomplete* set of filter atoms and expands the tunable parameter space, all while still requiring fewer parameters than LoRA. Additionally, we provide a simple technique to extend this method to linear layers, ensuring alignment with the characteristics in prior literature (Chen et al., 2023; Cheng et al., 2021b; Li et al., 2019; Miao et al., 2021a;b; Papyan et al., 2017; Qiu et al., 2018a; Wang et al., 2021a;b; Wang, 2023). Our method is illustrated in Figure 2.

We demonstrate the effectiveness of our approach on both discriminative and generative tasks with ResNet50 (He et al., 2016), ConvNeXt (Liu et al., 2022) and Stable Diffusion (Rombach et al., 2022). We summarize our contributions as follows,

- We propose a method by adapting only filter subspace elements (filter atoms), with a few hundred parameters, to achieve significantly parameter-efficient fine-tuning.
- We observe that maintaining fixed atom coefficients plays a crucial role in preserving the generalization capability of large models.
- We further demonstrate a simple way to expand the number of tunable parameters in filter subspace by recursively decomposing each filter atom over another set of filter atoms, which extends the parameter space for tuning.
- We conduct extensive experiments demonstrating the efficacy of our approach on discriminative and generative tasks for fine-tuning large models.

## 2 PRELIMINARY

### 2.1 LOW-RANK ADAPTATION FOR FINE-TUNING

LoRA (Hu et al., 2021) introduces trainable low-rank matrices into layers to approximate weight updates. For a pre-trained weight matrix $\mathbf{W} \in \mathbb{R}^{c' \times c}$, LoRA represents the weight update with a low-rank decomposition

$$\mathbf{W} + \Delta\mathbf{W} = \mathbf{W} + \mathbf{W}_{down}\mathbf{W}_{up},$$

where $\mathbf{W}_{down} \in \mathbb{R}^{c' \times r}$ and $\mathbf{W}_{up} \in \mathbb{R}^{r \times c}$ are tunable parameters, $r$ is the intrinsic rank of weight updates, $d'$ and $d$ are the size of inputs and outputs. $r$ is usually much smaller than $c'$ and $c$. A learnable scalar hyperparameter $\alpha$ scales the weight updates, which leads to $\mathbf{W} + \alpha\mathbf{W}_{down}\mathbf{W}_{up}$.

### 2.2 SPARSE CODING AND MATRIX FACTORIZATION

Sparse coding attempts to find the representation of input $\mathbf{w}$ with respect to a dictionary of $m$ atoms $\{\mathbf{d}_l\}_{l=1}^m$ with the fewest number of coefficients $\alpha^l$, *i.e.*, $\mathbf{w} = \sum_{l=1}^m \alpha^l \mathbf{d}_l$. The objective of a sparse coding problem can be written as

$$\min_{\alpha^{i,l}, \mathbf{d}_l} \sum_j \|\mathbf{w}^i - \sum_i \alpha^{i,l}\mathbf{d}_l\|_2^2 + \lambda \sum_i \|\alpha^{i,l}\|_1,$$

where $\|\cdot\|_1$ is the $L_1$ norm, and $\lambda$ is a Lagrange multiplier.

It can be further expressed in a tensor form,

$$\min_{\boldsymbol{\alpha}, \mathbf{D}} \|\mathcal{F} - \boldsymbol{\alpha} \times \mathbf{D}\|_F^2 + \lambda\|\boldsymbol{\alpha}\|_{1,1}, \tag{1}$$

where $\mathcal{F} \in \mathbb{R}^{c' \times c \times k' \times k}$ is the input tensor, $\boldsymbol{\alpha} \in \mathbb{R}^{c' \times c \times m}$ is the tensor of coefficients and $\mathbf{D} \in \mathbb{R}^{m \times k' \times k}$ is a tensor of basis. Several algorithms have been developed to solve (1), such as fast iterative soft-thresholding algorithm (FISTA) (Beck & Teboulle, 2009), matching pursuit (Mallat & Zhang, 1993), and least absolute shrinkage and selection operator (LASSO) (Santosa & Symes, 1986). We can interpret the first term in (1) as finding a suitable tensor factorization for $\mathcal{F}$, while the second term serves as a penalty enforcing sparsity in the representation of $\mathcal{F}$.

## 3 METHODS

In this section, we decompose convolution filters over a small set of filter subspace elements, referred to as *fitler atoms*. This formulation enables a new model tuning method via filter subspace by solely adjusting filter atoms.

### 3.1 FORMULATION OF FILTER DECOMPOSITION

Our approach involves decomposing each convolutional layer $\mathcal{F}$ into two standard convolutional layers: a *filter atom* layer $\mathbf{D}$ that models filter subspace[2], and an *atom coefficient* layer $\boldsymbol{\alpha}$ with $1 \times 1$ filters that represent combination rules of filter atoms, as displayed in Figure 2 (a). This formulation is written as

$$\mathcal{F} = \boldsymbol{\alpha} \times \mathbf{D}, \tag{2}$$

where $\times$ is the tensor product.

$\mathcal{F} \in \mathbb{R}^{c' \times c \times k \times k}$ contains $c' \times c$ filters $\{\mathcal{F}^{i,j}\}_{i=1,j=1}^{i=c',j=c}$, where $c'$ and $c$ are the numbers of input and output channels, $k$ is the kernel size. $\mathbf{D} \in \mathbb{R}^{m \times k \times k}$ is a set of $m$ filter atoms, *i.e.*, $\mathbf{D} = \{\mathbf{d}_l\}_{l=1}^m$, where $\mathbf{d}_l \in \mathbb{R}^{k \times k}$. Each filter $\mathcal{F}^{i,j}$ is a linear combination of $m$ filter atoms, calculated by $\mathcal{F}^{i,j} = \sum_{l=1}^m \alpha^{i,j,l}\mathbf{d}_l$, $\alpha^{i,j,l}$ represents each element in $\boldsymbol{\alpha} \in \mathbb{R}^{c' \times c \times m}$.

Considering input features $\mathbf{X} \in \mathbb{R}^{c' \times h' \times w'}$ and output features $\mathbf{Z} \in \mathbb{R}^{c \times h \times w}$, where $h' \times w'$ and $h \times w$ represent the dimension of input features and output features. The convolution process now can be comprehended as the following two steps:

---

[2]The filter subspace is a span of $m$ filter atoms $\mathbf{D}$.

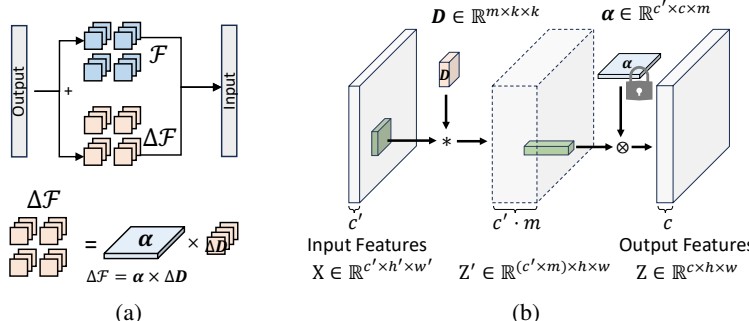

Figure 2: A filter subspace view of model tuning. (a) Each convolutional layer $\mathcal{F}$ is constructed as linear combinations of filter subspace elements, *i.e.*, *filter atoms* $\mathbf{D}$, using decomposition coefficients, *i.e.*, *atom coefficients* $\boldsymbol{\alpha}$. Hypothesizing variations across tasks can be reduced by bridging spatial discrepancies in the images, we propose to calibrate the pre-trained model by solely fine-tuning the spatial-only $\mathbf{D}$ while keeping the spatially-invariant channel weights $\boldsymbol{\alpha}$ fixed. (b) The convolution operation is represented in two stages: At the *spatial-only convolution* stage, each filter atom $\mathbf{D}$ is adapting to the target task, and then at the *cross-channel mixing* stage, $\mathbf{Z}'$ are combined into $\mathbf{Z}$ using fixed atom coefficients $\boldsymbol{\alpha}$, which is obtained from the pre-trained model. More details are provided in Section 3.

● **Spatial-only Convolution with D.** Each channel of the input features $\mathbf{X}$ convolves with each filter atom separately to produce intermediate features

$$\mathbf{Z}' = \mathbf{D} * \mathbf{X},$$

where $\mathbf{Z}' \in \mathbb{R}^{(c' \times m) \times h \times w}$, $*$ is the convolution operation.

For each input feature channel $\mathbf{X}^i$ and filter atom $\mathbf{d}_l \in \mathbb{R}^{k \times k}$, we have $\mathbf{Z}'^{i,l} = \mathbf{X}^i * \mathbf{d}_l$, where $\mathbf{Z}'^{i,l} \in \mathbb{R}^{h \times w}$, $\mathbf{X}^i \in \mathbb{R}^{h' \times w'}$ are one feature channel in $\mathbf{Z}', \mathbf{X}$, and $\mathbf{d}_l$ is one filter atom in $\mathbf{D} = \{\mathbf{d}_l\}_{l=1}^m$.

This process leads to the generation of $m$ distinct intermediate output channels for each input channel, which is illustrated in Figure 2 (b). In this step, filter atoms focus only on handling the spatial information of input features, and cross-channel mixing is postponed to the next step.

● **Cross-channel Mixing with $\boldsymbol{\alpha}$.** Subsequently, atom coefficients weigh and linearly combine the intermediate features to produce output features

$$\mathbf{Z} = \boldsymbol{\alpha} \times \mathbf{Z}'.$$

Each output feature channel $\mathbf{Z}^j \in \mathbb{R}^{h \times w}$ is linearly combined from $c' \times m$ intermediate feature channels $\mathbf{Z}'^{i,l} \in \mathbb{R}^{h \times w}$ with the coefficient $\{\alpha^{i,j,l}\}_{i=1,l=1}^{i=c',l=m}$, which is, $\mathbf{Z}^j = \sum_{i=1}^{c'} \sum_{l=1}^{m} \alpha^{i,j,l} \cdot \mathbf{Z}'^{i,l}$. Here, $\alpha^{i,j,l}$ represents each element in atom coefficients $\boldsymbol{\alpha} \in \mathbb{R}^{c' \times c \times m}$.

The spatially invariant channel weights, atom coefficients $\boldsymbol{\alpha}$, serve as operators for channel mixing, functioning as distinct combination rules that construct the output features from the elemental feature maps generated by the filter atoms. During the model tuning, $\boldsymbol{\alpha}$ is obtained from the pre-trained model and remains unchanged, while only filter atoms $\mathbf{D}$ adapt to the target task.

**Summary.** The two-step convolution operation explains different functionalities of filter atoms $\mathbf{D}$ and atom coefficients $\boldsymbol{\alpha}$, which is, $\mathbf{D}$ only contribute to spatial convolution and $\boldsymbol{\alpha}$ only perform cross-channel mixing. In practice, the convolution operation is still performed as one layer, without generating intermediate features, to avoid memory cost. In the fine-tuning process, we solely adjust $\mathbf{D}$, which contains a small set of number of parameters, $k \times k \ll c' \times c$, thereby facilitating parameter-efficient fine-tuning.

## 3.2 OVERCOMPLETE FILTER ATOMS

The parameters of filter atoms are extremely small compared with overall model parameters. For instance, the filter atoms constitute a mere $0.004\%$ of the total parameters in ResNet50 (He et al.,

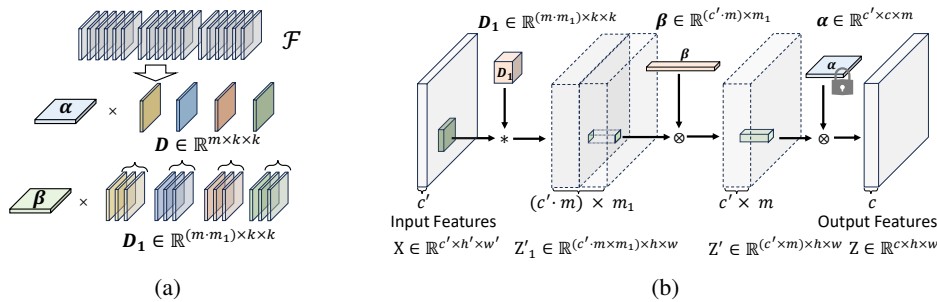

(a)                                                          (b)

Figure 3: (a) The process of constructing overcomplete filter atoms: One filter atom in $\mathbf{D}$ is presented as a linear combination of $m_1$ filter atoms in $\mathbf{D}_1$. (b) The convolution operation is represented in three stages: At the spatial-only convolution stage, a set of overcomplete filter atoms $\mathbf{D}_1$ is adapting to the target task and generates $m \cdot m_1$ feature channels by convolving with each input channel. At the *intra-channel mixing* stage, every group of $m_1$ features is linearly combined by coefficients $\boldsymbol{\beta}$ with *no cross-channel mixing*. At the cross-channel mixing stage, $\mathbf{Z}'$ are combined into $\mathbf{Z}$ using fixed atom coefficients $\boldsymbol{\alpha}$, which is obtained from the pre-trained model.

2016). To fully explore the potential of filter subspace fine-tuning, we show next a simple way to construct a set of overcomplete[3] filter atoms by recursively applying the above decomposition to each filter atom, to expand the parameter space for fine-tuning as needed.

Specifically, each filter atom $\mathbf{d}_l$ can be further decomposed over $m_1$ number of filter atoms in $\mathbf{D}_{1,l} = \{\mathbf{d}_{1,l}^j\}_{j=1}^{m_1}$ with its corresponding coefficients $\boldsymbol{\beta}^i = [\beta^{i,1}, \cdots, \beta^{i,m_1}] \in \mathbb{R}^{m_1}$, *i.e.*, $\mathbf{d}_l = \sum_{j=1}^{m_1} \beta^{i,j} \mathbf{d}_{1,l}^j$. Therefore, $\mathbf{D}_1 \in \mathbb{R}^{(m \cdot m_1) \times k \times k}$ contains $m \cdot m_1$ filter atoms, leading to a overcomplete set. The recursive decomposition process is illustrated in Figure 3 (a).

Considering input $\mathbf{X} \in \mathbb{R}^{c' \times h' \times w'}$ and output $\mathbf{Z} \in \mathbb{R}^{c \times h \times w}$, convolution using the overcomplete filter atoms can be comprehended as three steps. While the spatial-only convolution with $\mathbf{D}$ and cross-channel mixing with $\boldsymbol{\alpha}$ steps are the same as before, it linearly combines the features channels corresponding to $m_1$ decomposed filter atoms in *intra-channel mixing* step.

• **Intra-channel Mixing with $\boldsymbol{\beta}$.** After convolving $\mathbf{X}$ and $\mathbf{D}_1$ to get $\mathbf{Z}'_1 \in \mathbb{R}^{(c' \cdot m \times m_1) \times h \times w}$, the intermediate feature map $\mathbf{Z}' \in \mathbb{R}^{(c' \cdot m) \times h \times w}$ is obtained by linearly combining feature channels within the corresponding input channel via coefficients $\boldsymbol{\beta}$,

$$\mathbf{Z}' = \boldsymbol{\beta} \times \mathbf{Z}'_1.$$

Each feature channel $\mathbf{Z}'^i$ is calculated by $\mathbf{Z}'^i = \sum_{j=1}^{m_1} \beta^{i,j} \mathbf{Z}_1'^{i,j}$, where $\beta^{i,j}$ is one element in $\boldsymbol{\beta}$, and $\mathbf{Z}_1'^{i,j} \in \mathbb{R}^{h \times w}$ is one feature channel in $\mathbf{Z}'_1$. This phase exclusively blends the feature channels within the same input channel, avoiding any mixing across different channels. The whole process is illustrated in Figure 3 (b). To adapt the model to downstream tasks, we fine-tune both $\boldsymbol{\beta}$ and $\mathbf{D}_1$, which contain more tunable parameters than $\mathbf{D}$.

This method can be readily adapted to $1 \times 1$ convolutional and linear layers as well. We start by applying Kronecker decomposition on a linear layer $\mathbf{W} = \mathbf{A} \otimes \mathbf{B}$, where $\mathbf{A} \in \mathbb{R}^{\frac{c'}{k_c'} \times \frac{c}{k_c}}$, $\mathbf{B} \in \mathbb{R}^{k_c' \times k_c}$, $\mathbf{W} \in \mathbb{R}^{c' \times c}$, $\otimes$ is the Kronecker product. This approach has been explored by recent literature on fine-tuning (Edalati et al., 2022; YEH et al., 2023). The idea is to represent a matrix as multiple blocks, where each block comes from the same basis $\mathbf{B}$ but varied by different constant $A_{ij}$. We then introduce $m_c$ pairs of $\mathbf{A}$ and $\mathbf{B}$ to reconstruct $\mathbf{W}$, $\mathbf{W} = \sum_{i=1}^{m_c} \mathbf{A}_i \otimes \mathbf{B}_i$, such that each block in $\mathbf{W}$ is represented as a linear combination of all $\mathbf{B}_i$.

### 3.3   DECOMPOSITION OF LINEAR LAYERS

---

[3]As the size of filters is $k \times k$, $m = k^2$ independent filter atoms is *complete* since every filter can be linearly combined by these filter atoms. As the number of filter atoms is larger than $k^2$, it becomes *overcomplete*. Overcompleteness can potentially bring a more stable fine-tuning, and in our case, expand the number of parameters for adapting to downstream tasks.

We re-write the above formulation as the notation with coefficients and atoms such that $\boldsymbol{\alpha}_c = \{\mathbf{A}_i\}_{i=1}^{m_c}$ and $\mathbf{D}_c = \{\mathbf{B}_i\}_{i=1}^{m_c}$. Therefore, $\mathbf{W}$ is constructed from $m_c$ atoms in $\mathbf{D}_c \in \mathbb{R}^{m_c \times k'_c \times k_c}$ by the combination using coefficients $\boldsymbol{\alpha}_c \in \mathbb{R}^{\frac{c'}{k'_c} \times \frac{c}{k_c} \times m_c}$, which is, $\mathbf{W} = \boldsymbol{\alpha}_c \times \mathbf{D}_c$. This formulation is mathematically equal to the decomposition filters as filter atoms and atom coefficients. Similarly, the $1 \times 1$ convolutional layer $\mathcal{F}_c \in \mathbb{R}^{c' \times c \times 1 \times 1}$ can be formulated as $\boldsymbol{\alpha}_c$ and $\mathbf{D}_c$ with the same process,

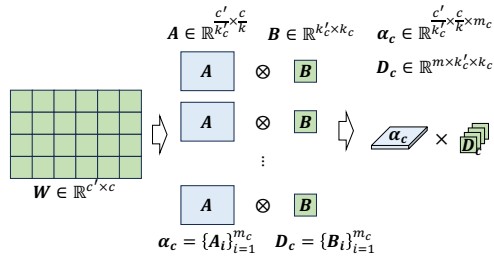

Figure 4: Formulate the linear layer $\mathbf{W}$ as a combination of atoms, $\mathbf{W} = \boldsymbol{\alpha}_c \times \mathbf{D}_c$.

$$\mathcal{F}_c = \boldsymbol{\alpha}_c \times \mathbf{D}_c. \tag{3}$$

The procedure of formulating linear layers as coefficients and atoms is illustrated in Figure 4. During the model tuning, $\boldsymbol{\alpha}_c$ is obtained from the pre-trained model and remains unchanged, while only filter atoms $\mathbf{D}_c$ adapt to the target task.

## 3.4 PARAMETER EFFICIENT FINE-TUNING

Building on our filter subspace formulation, we investigate various parameter-efficient fine-tuning strategies for tuning the pre-trained model. For convolutional filters $\mathcal{F}$, we directly use sparse coding (1) to decompose them as atom coefficients and filter atoms $\boldsymbol{\alpha} \times \mathbf{D}$. For linear layers or $1 \times 1$ convolution layers, we first formulate them as $\mathcal{F}_c$ (3), and then decompose them as coefficients $\boldsymbol{\alpha}_c$ and atoms $\mathbf{D}_c$ using (1). For fine-tuning the models, we only adapt $\mathbf{D}$ or $\mathbf{D}_c$, while keeping $\boldsymbol{\alpha}$ or $\boldsymbol{\alpha}_c$ fixed. When more parameters are needed, we represent $\mathbf{D}$ as $\boldsymbol{\beta}$ and $\mathbf{D}_1$ to incorporate overcomplete filter atoms. We initialize $\mathbf{D}_1$ by simply repeating each filter atom in $\mathbf{D}$ for $m_1$ times and initialize every element in $\boldsymbol{\beta}$ with the value $1/m_1$. This enables us to fine-tune both $\boldsymbol{\beta}$ and $\mathbf{D}_1$.

In this study, we investigate three strategies that offer a range of tunable parameters from a small set to a slightly larger number: fine-tuning (1) only $\mathbf{D}$, (2) $\mathbf{D}$ and $\mathbf{D}_c$, and (3) $\boldsymbol{\beta}$, $\mathbf{D}_1$ and $\mathbf{D}_c$. Downstream tasks are modeled by their updates $\Delta \mathbf{D}$, $\Delta \boldsymbol{\beta}$, $\Delta \mathbf{D}_1$ and $\Delta \mathbf{D}_c$. In the Appendix B, we present an analysis of the decomposition complexity and a comparison of the number of parameters with baseline methods.

## 4 EXPERIMENTS

In this section, we begin with studying the effectiveness of our method across various configurations to determine the most suitable application scenario for each configuration. Subsequently, we demonstrate that fine-tuning only filter atoms requires far fewer parameters while preserving the capacity of pre-trained models, compared with baseline methods in the contexts of discriminative and generative tasks.

### 4.1 EXPERIMENTAL SETTINGS

**Datasets.** Our experimental evaluations are mainly conducted on the Visual Task Adaptation Benchmark (VTAB) (Zhai et al., 2019), which contains 19 distinct visual recognition tasks sourced from 16 datasets. As a subset of VTAB, VTAB-1k comprises a mere $1,000$ labeled training examples in each dataset.

**Models.** For the validation experiment, we choose ResNet50 (He et al., 2016) pre-trained on ImageNet-1K. For discriminative tasks, we choose the convolution-based model, ConvNeXt-B (Liu et al., 2022) pre-trained on ImageNet-21K as the initialization for fine-tuning. For generative tasks, we choose Stable Diffusion (Rombach et al., 2022) which contains a downsampling-factor 8 autoencoder with an 860M UNet and CLIP ViT-L/14 as text encoder for the diffusion model. The model is pre-trained on the LAION dataset (Schuhmann et al., 2022), which contains over 5 billion image-text pairs. More details of pre-trained models are listed in the Appendix A.

Table 1: Comparison among different configurations of our method.

| | Linear Probe | LoRA | Full Finetuning | Fine-tune $\mathbf{D}$ | | |
| --- | --- | --- | --- | --- | --- | --- |
| | | | | $m=6$ | $m=9$ | $m=12$ |
| Accuracy $\uparrow$ | 55.4 | 78.6 | 83.3 | 66.25 | 66.86 | 68.68 |
| Param. (M) $\downarrow$ | 0.2 | 2 | 25.6 | **0.21** | **0.21** | **0.21** |

| | Fine-tune $\boldsymbol{\beta}$ and $\mathbf{D}_1$ | | | Fine-tune $\mathbf{D}$ and $\mathbf{D}_c$ | | FT $\boldsymbol{\beta}$, $\mathbf{D}_1$ and $\mathbf{D}_c$ |
| --- | --- | --- | --- | --- | --- | --- |
| | $m_1=3$ | $m_1=4$ | $m_1=5$ | $k_c=2$ | $k_c=4$ | $m_1=4, k_c=4$ |
| Accuracy $\uparrow$ | 78.69 | 78.7 | 78.9 | 75.03 | 79.83 | **81.8** |
| Param. (M) $\downarrow$ | 0.82 | 1 | 1.2 | 0.23 | 0.62 | 2.1 |

## 4.2 Validation Experiments

In this section, we study the performance of our approach across various configurations.

**Implementation details.** We employ ResNet50 (He et al., 2016) pre-trained on ImageNet-1K (Russakovsky et al., 2015) and fine-tune it on CIFAR-100 (Krizhevsky & Hinton, 2009) for 50 epochs. We utilize the Adam optimizer (Kingma & Ba, 2015) with a learning rate of 0.001 and a weight decay of $1 \times 10^{-4}$ and a batch size of 256.

We compare different configures of our methods with the following experimental settings and $\boldsymbol{\alpha}$, $\boldsymbol{\alpha}_c$ are always fixed: (1) When fine-tuning $\mathbf{D}$, we experiment with different numbers of filter atoms $m$ in the range of $[6, 9, 12]$. (2) When fine-tuning $\mathbf{D}$ and $\mathbf{D}_c$, we choose $(k_c', k_c) \in \{(2, 2), (4, 4)\}$, and the number of filter atoms is $m = 9$. (3) When fine-tuning $\mathbf{D}_1$ and $\boldsymbol{\beta}$, we choose $m = 9$, and $m_1$ in the range of $[3, 4, 5]$. (4) When fine-tuning $\mathbf{D}_c$, $\mathbf{D}_1$ and $\boldsymbol{\beta}$, we choose $m = 9$, $m_1 = 3$ and $(k_c', k_c) = (2, 2)$.

**The effectiveness of filter subspace.** The results of all configurations are displayed in Table 1, with $m = 9$ as an example, and $m = 6, 12$ are detailed in the Appendix C. (1) While fine-tuning $\mathbf{D}$ with fixed $\boldsymbol{\alpha}$, our approach requires a negligible increase in the number of parameters, amounting to only 0.5% additional parameters, but it achieves an almost 20% improvement in accuracy when compared to linear probing. (2) Fine-tuning $\boldsymbol{\beta}$ and $\mathbf{D}_1$ further improves accuracy from 66.8% to 78.7% compared to fine-tuning $\mathbf{D}$ alone. It validates that overcomplete filter atoms provide more capacities for model tuning. (3) Fine-tuning $\mathbf{D}$ and $\mathbf{D}_c$ results in significant improvements compared to fine-tuning $\mathbf{D}$ alone. For example, with $k_c = 2$, fine-tuning 0.02M additional parameters improves accuracy from 66.86% to 75.03%. (4) Fine-tuning $\mathbf{D}_c$, $\mathbf{D}_1$, and $\boldsymbol{\beta}$ achieves the highest accuracy but it also requires the greatest number of parameters among all of our configurations.

**Discussion.** Fine-tuning the filter atoms $\mathbf{D}$ or atoms $\mathbf{D}_c$ enhances accuracy while keeping the parameter increase minimal, making it highly appropriate for scenarios where the number of parameters is a critical consideration. However, increasing the number of atoms does not easily yield further accuracy improvements. To achieve additional accuracy gains, incorporating the set of overcomplete filter atoms $\mathbf{D}_1$ and their coefficients $\boldsymbol{\beta}$ can be effective, but at the cost of an increase in parameters.

## 4.3 Generative Tasks

**Baselines.** We compare our method to 8 baseline fine-tuning approaches: (i) Full fine-tuning, which entails updating all model parameters during the fine-tuning process; (ii) LoRA (Hu et al., 2021), involving the introduction of a low-rank structure of accumulated gradient update by decomposing it as up-projection and down-projection. (iii) LoHa (YEH et al., 2023) utilizes the Hadamard product across two sets of low-rank decompositions to elevate the rank of the resultant matrix and reduce the approximation error. (iv) LoKr (YEH et al., 2023) introduces the Kronecker product for matrix decomposition to reduce the tunable parameters. (v) BitFit (Zaken et al., 2022) fine-tunes the bias term of each layer. (vi) DiffFit (Xie et al., 2023) fine-tunes the bias term, as well as the layer norm and the scale factor of each layer. (vii) OFT and COFT (Qiu et al., 2023) adapts the diagonal blocks of weight matrices to achieve orthogonal fine-tuning.

**Implementation details.** We fine-tune Stable Diffusion using the AdamW optimizer (Loshchilov & Hutter, 2018) with a learning rate of $5 \times 10^{-6}$ for full parameter fine-tuning, and $1 \times 10^{-3}$ for all other parameter-efficient fine-tuning methods. For few-shot learning, we adopt 30 different customized concepts from Dreambooth (Ruiz et al., 2023) and fine-tune the model on $4 \sim 9$ images. We utilize 25 different text prompts following (Qiu et al., 2023; Ruiz et al., 2023) and generate

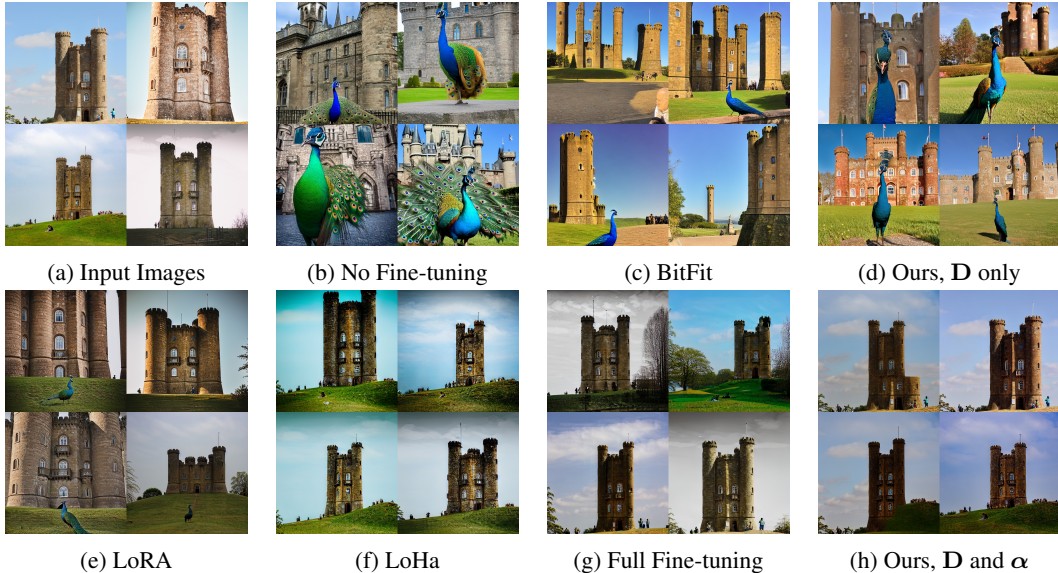

| (a) Input Images | (b) No Fine-tuning | (c) BitFit | (d) Ours, $\mathbf{D}$ only |
|:---:|:---:|:---:|:---:|

| (e) LoRA | (f) LoHa | (g) Full Fine-tuning | (h) Ours, $\mathbf{D}$ and $\boldsymbol{\alpha}$ |
|:---:|:---:|:---:|:---:|

Figure 5: Fine-tune Stable Diffusion (Rombach et al., 2022) to learn the concept $\langle$castle$\rangle$ from (a) and generate images using text prompt: "A peacock in front of the $\langle$castle$\rangle$". (b) Images generated by the pre-trained model, without any fine-tuning, align well with the text prompt but fail to match the target concept. (d) When only fine-tuning the filter atoms $\mathbf{D}$, our approach achieves good alignment with the text prompt and the target concept. (h) However, fine-tuning the spatially-invariant channel weights, $i.e.$, atom coefficients $\boldsymbol{\alpha}$, results in generated images only align with the target concept, compromising the capability of the pre-trained model of producing diverse images that align with the text prompt. This issue is also observed in (e-g).

Table 2: Evaluate different approaches in learning the customized concept.

| | Pre-trained | Full FT | LoRA | LoHa | LoKr | OFT | COFT | DiffFit | BitFit | $\mathbf{C}_1$ | $\mathbf{C}_2$ | $\mathbf{C}_3$ | $\mathbf{C}_4$ |
|---|---|---|---|---|---|---|---|---|---|---|---|---|---|
| Fidelity ↑ | 0.144 | 0.711 | 0.697 | 0.693 | 0.693 | 0.656 | 0.652 | 0.622 | 0.571 | 0.594 | 0.652 | 0.707 | **0.716** |
| Diversity ↑ | 42.88 | 4.01 | 4.84 | 3.96 | 5.14 | 5.86 | 5.92 | 7.22 | 10.08 | **20.42** | 9.37 | 6.92 | 3.17 |
| T2I Alignment ↑ | 0.332 | 0.234 | 0.232 | 0.216 | 0.238 | 0.267 | 0.264 | 0.268 | 0.277 | **0.301** | 0.279 | 0.236 | 0.201 |
| Param. (M) ↓ | - | 860 | 22.67 | 8.47 | 1.06 | 11.75 | 11.75 | 0.58 | 0.34 | **0.05** | 0.75 | 2.39 | 587 |

images with a shape of $512 \times 512$ for each concept using these prompts. We follow the method in YEH et al. (2023) and fine-tune the model for 1000 steps, and evaluate the fidelity, diversity, and text-to-image alignment of generated images.

We assess the fidelity as the average cosine similarity between DINOv2 embeddings (Oquab et al., 2023) of the generated and dataset images. The diversity of generated images is measured by the Vendi score (Friedman & Dieng, 2022) calculated with the DINOv2 embeddings. The alignment between generated images and corresponding prompts is measured via average cosine similarity in the CLIP feature space (Radford et al., 2021).

For the VTAB dataset, we fine-tune $\boldsymbol{\beta}$, $\mathbf{D}_1$, and $\mathbf{D}_c$ of the model for 5000 steps with the image shape of $256 \times 256$ and utilize Frechet Inception Distance (FID) (Heusel et al., 2017) as a quantitative metric for evaluation. To calculate FID, we generate 20,000 images from each checkpoint and compare them with images from the corresponding dataset. In cases where the dataset contains more than 20,000 images, we sample 20,000 images for comparison. See the Appendix A for details of experimental settings.

**Few-shot generative transfer learning.** We report the evaluation of all methods in Table 2. Our method with different configurations is denoted as $C_1 - C_4$. (1) $C_1$ and $C_2$ represent fine-tuning $\mathbf{D}$ and $\mathbf{D}_c$ of the model with $(m, k_c) \in \{(6, 4), (9, 16)\}$. (2) $C_3$ fine-tunes $\boldsymbol{\beta}$, $\mathbf{D}_1$ and $\mathbf{D}_c$, with $(m, k_c, m_1) = (9, 16, 3)$. (3) $C_4$ is similar to $C_2$ but it fine-tunes $\boldsymbol{\alpha}$ together with $\mathbf{D}$ and $\mathbf{D}_c$.

Compared with other baseline methods, our method with configuration $C_1 - C_3$ generate diverse images that are well-aligned with the text prompt. It means they preserve the capability of the pre-

Table 3: FIDs (lower the better) of image generation models on VTAB benchmark with Stable Diffusion pre-trained on LAION.

| | Natural | | | | | | | Specialized | | | | Structured | | | | | | |
| --- | --- | --- | --- | --- | --- | --- | --- | --- | --- | --- | --- | --- | --- | --- | --- | --- | --- | --- |
| | Caltech101 | CIFAR100 | DTD | Flowers102 | Pets | SVHN | Sun397 | Patch Camelyon | EuroSAT | Resisc45 | Retinopathy | Clevr | DMLab | KITTI | dSprites | SmallNORB | Mean | Params. (M) |
| No fine-tuning | 52.0 | 113.3 | 74.5 | 45.5 | 117.1 | 212.2 | 28.8 | 258.8 | 186.2 | 144.2 | 307.7 | 271.3 | 225.0 | 288.5 | 373.9 | 266.7 | 179.1 | - |
| Full fine-tuning | 39.0 | 33.8 | 46 | 42.9 | 32.6 | 93.2 | 17.0 | 107.6 | 144.5 | 54.3 | 69.0 | 25.8 | 30.3 | 92.9 | 75.8 | 75.6 | 61.3 | 860 |
| LoRA (Hu et al., 2021) | 36.8 | 42.4 | 40.6 | 45.3 | 30.8 | 120.9 | 17.7 | 101.2 | 81.9 | 61.6 | 69.5 | 33.6 | 32.1 | 64.4 | 69.4 | 71.6 | 57.5 | 22.67 |
| LoHa (YEH et al., 2023) | 33.8 | 40.5 | 41.6 | 41.5 | 33 | 140.9 | 17.3 | 111.4 | 73.7 | 60.4 | 66.5 | 32.1 | 37.9 | 64.5 | 73.3 | 65.7 | 58.4 | 33.86 |
| LoKr (YEH et al., 2023) | 37.7 | 43.5 | 43.2 | 50.6 | 41.1 | 144.2 | 21.4 | 95.4 | 70.5 | 65.5 | 79.3 | 33.9 | 44.8 | 64.1 | 79.4 | 74.4 | 61.8 | 2.12 |
| Ours | 38.6 | 32.4 | 43.9 | 38.5 | 30.6 | 96.8 | 17.2 | 109.7 | 64.1 | 62.4 | 60.6 | 18.7 | 41.9 | 69.5 | 73.2 | 76.5 | 54.7 | 1.11 |

trained model while learning the new concept. However, $C_4$ fine-tunes the atom coefficients $\alpha$, resulting in the model overfitting to the target concept, which is reflected in a notably high fidelity score, while compromising the model's ability to generate images aligned with the text prompt. This observation suggests that maintaining the spatially invariant channel weights $\alpha$ helps prevent overfitting when fine-tuning pre-trained models to downstream tasks. Figure 1 illustrates visual examples of learning the concept "castle" from CustomConcept101 dataset (Kumari et al., 2023).

Methods like LoRA (Hu et al., 2021) or full fine-tuning potentially update these $\alpha$, thus, they lead to lower diversity and text-to-image alignment in generated images. In contrast, BitFit (Zaken et al., 2022) and DiffFit (Xie et al., 2023) mostly fine-tune the bias, leaving $\alpha$ fixed, thus, they have a higher diversity and text-to-image alignment than LoRA. However, they also keep the spatial operation $\mathbf{D}$ unchanged, resulting in a lower fidelity score compared with $C_2$. More results can be found in Appendix C.

**Performance comparisons on generative transfer learning.** We report FIDs of models trained and evaluated on VTAB tasks in Table 3. In contrast to full parameter fine-tuning and LoRA, our approach attains the lowest FID scores ($54.7$ v.s. $57.5$) while employing the least number of fine-tuning parameters ($1.11\text{M}$ v.s. $22.67\text{M}$). Despite fine-tuning only $0.13\%$ of the total model parameters, our method effectively tailors pre-trained Stable Diffusion to align it with the desired target distribution.

## 4.4 DISCRIMINATIVE TASKS

In this section, we apply our method to the discriminative task, namely the classification on VTAB-1k (Zhai et al., 2019). We compare our method to 4 baseline fine-tuning approaches: (i) Full fine-tuning, (ii) Linear probing, (iii) BitFit (Zaken et al., 2022), and (iv) LoRA (Hu et al., 2021).

**Implementation details.** Images are resized to $224 \times 224$, following the default settings in VTAB (Zhai et al., 2019). We employ the AdamW (Loshchilov & Hutter, 2018) optimizer to fine-tune models for 100 epochs. The cosine decay strategy is adopted for the learning rate schedule, and the linear warm-up is used in the first 10 epochs. In this experiment, we fine-tune $\mathbf{D}$ and $\mathbf{D}_c$ while keeping $\alpha$ and $\alpha_c$ fixed, as this configuration delivers adequate accuracy without increasing parameters.

**Performance comparisons on few-shot transfer learning.** We compare the performance of our approach and other baseline methods, and the results on VTAB-1k are shown in Table 4. In these tables, the bold font shows the best accuracy of all methods and the underlined font shows the second best accuracy. Our method outperforms other parameter-efficient fine-tuning methods and even outperforms full fine-tuning. Specifically, our method obtains $6\%$ improvement in accuracy compared to LoRA on the VTAB-1k benchmark while utilizing significantly fewer trainable parameters ($0.45\text{M}$ v.s. $17.4\text{M}$). The Appendix C also includes the experimental results for ViT-B/16.

## 5 RELATED WORKS

**Pre-training and Fine-tuning.** The standard practice of pre-training and fine-tuning (He et al., 2016; Huang et al., 2017; Tan & Le, 2019; Xie et al., 2017) entails models initially undergoing pre-training on datasets such as ImageNet-21K, BookCorpus, and Common Crawl (Russakovsky et al., 2015; Raffel et al., 2020; Zhu et al., 2015). Subsequently, these models are fine-tuned to enhance their convergence and performance on specific tasks (He et al., 2019). In the realm

Table 4: Performance comparisons on the VTAB-1k benchmark with ConvNeXt models pre-trained on ImageNet-21K.

| | Natural | | | | | | | Specialized | | | | Structured | | | | | | | | | |
|---|---|---|---|---|---|---|---|---|---|---|---|---|---|---|---|---|---|---|---|---|---|
| | Caltech101 | CIFAR100 | DTD | Flowers102 | Pets | SVHN | Sun397 | Patch Camelyon | EuroSAT | Resisc45 | Retinopathy | Clevr/count | Clevr/distance | DMLab | KITTI | dSprites/loc | dSprites/ori | SmallNORB/azi | SmallNORB/ele | Mean | Params. (M) |
| Full fine-tuning | **94.9** | 64.2 | 73.6 | 99.5 | 90.8 | 89.6 | 37.7 | **86.6** | 85.1 | 85.9 | 73.6 | **73.3** | 61.3 | **52.1** | **83.1** | 86.8 | **61.1** | 32.7 | **38.8** | 72.14 | 87.67 |
| Linear Probing | 92.3 | 65.8 | 76.8 | 99.3 | 92.7 | 50.5 | 55.8 | 84.0 | 92.7 | 82.5 | 74.7 | 46.1 | 38.5 | 41.1 | 66.3 | 24.2 | 35.4 | 18.4 | 26.0 | 61.21 | 0.11 |
| BitFit (Zaken et al., 2022) | 94.5 | 71.6 | 76.7 | 99.4 | 93.0 | 85.7 | 57.2 | 86.4 | **94.0** | **86.4** | 74.3 | 67.8 | 57.2 | 49.8 | 80.5 | 77.7 | 59.1 | 30.4 | 22.0 | 71.77 | 0.24 |
| LoRA (Hu et al., 2021) | 94.3 | 51.7 | 61.4 | 88.1 | 69.8 | **91.2** | 38.1 | 74.5 | 91.9 | 81.4 | 73.6 | 60.8 | 62 | 50.3 | 80.1 | **96.3** | 56.3 | **39.3** | 21.9 | 67.53 | 17.4 |
| Ours | 94.8 | **71.7** | **76.9** | **99.6** | **93.1** | 87.1 | **57.5** | 85.1 | **94.6** | **87.6** | **74.8** | 70.9 | **62.8** | 50.3 | 82.7 | 89.4 | 60.4 | 31.2 | 29 | **73.59** | 0.45 |

of parameter-efficient fine-tuning (Zhou et al., 2022), various approaches have been proposed. LoRA (Hu et al., 2021) fine-tunes lower-rank matrices at each layer to represent weight updates. The adapter (Houlsby et al., 2019) approach inserts small modules between layers and reduces parameters by only tuning these adapters (Chen et al., 2022; Karimi Mahabadi et al., 2021; Li & Liang, 2021; Zaken et al., 2022). Visual prompt tuning (VPT) (Jia et al., 2022; Sohn et al., 2023) has introduced a limited number of learnable parameters for optimization while keeping the backbone frozen. SSF (Lian et al., 2022) proposes scaling and shifting deep features extracted by a pre-trained model.

**Model Architectures.** Compared with transformer-based models (Dosovitskiy et al., 2020; Liu et al., 2021; Touvron et al., 2021; Yu et al., 2022), convolution has been used for a long time as the main module to extract the image features in computer vision tasks. With an inductive prior, convolution-based models require fewer training images and computation resources to achieve good generalization. Convolution-based architectures have been largely studied (He et al., 2016; Liu et al., 2022; Simonyan & Zisserman, 2015) and have found multiple applications, such as feature extracting (Razavi et al., 2019), image generation (Karras et al., 2020; Song et al., 2021), super-resoluton (Wang et al., 2020), and et cetera. Numerous studies explore the integration of convolutional techniques with vision transformers to enhance their performance (Guo et al., 2022; Raghu et al., 2021). Parameter-efficient fine-tuning in downstream tasks is crucial and requires further examinations when utilizing pre-trained large-scale convolution-based models.

**Discriminative and Generative Tasks.** Discriminative and generative tasks are fundamental in machine learning. Discriminative models (Hao et al., 2020; He et al., 2016; Padilla et al., 2020; Zou et al., 2023) are designed to distinguish between data instances, while generative models (Karras et al., 2020; Razavi et al., 2019; Song et al., 2021; Wang et al., 2020) are employed to create new data instances. Discriminative models have been applied to image classifications (He et al., 2016; Liu et al., 2022; Simonyan & Zisserman, 2015), object detection (Padilla et al., 2020; Zou et al., 2023), and semantic segmentation (Hao et al., 2020). Generative models have been extensively studied for image synthesis, including variational autoencoder (Kingma et al., 2021; Razavi et al., 2019; Vahdat & Kautz, 2020; Van Den Oord et al., 2017), diffusion (Dhariwal & Nichol, 2021; Kim et al., 2024; Rombach et al., 2022; Song et al., 2021), and autoregressive models (Parmar et al., 2018; Van den Oord et al., 2016; Van Den Oord et al., 2016). In this study, our primary focus is on implementing parameter-efficient fine-tuning techniques for two tasks: image classification using ConvNeXt (Liu et al., 2022) and image synthesis employing Stable Diffusion (Rombach et al., 2022).

## 6 CONCLUSION

In this work, we proposed the parameter-efficient fine-tuning method for large convolutional models by formulating the convolutional layers over the filter subspace. Fine-tuning filter atoms composed of a small number of parameters and keeping the atom coefficients unchanged, is notably efficient in terms of parameters. It successfully maintains the capabilities of pre-trained models while avoiding overfitting to downstream tasks. We then formulate a simple yet effective way to achieve an over-complete filter subspace by decomposing each filter atom over another set of filter atoms, thereby expanding the parameter space available for fine-tuning as needed. Our approach has demonstrated effectiveness in different configurations on both discriminate and generative tasks.

**Limitations.** Our method, which concentrates on tuning models within the filter subspace, is particularly advantageous for ConvNets. While it can be naturally extended to linear layers through appropriate mathematical formulations, the full potential of our approach when applied to linear layers remains under-explored.

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

# A  DETAILS OF EXPERIMENTS

## A.1  DETAILS OF TASKS AND DATASETS

**VTAB Dataset.**  VTAB dataset is uniquely challenging and well-suited for the evaluation of parameter-efficient tuning methods in the context of few-shot knowledge transfer. VTAB-1k encompasses a diverse range of image domains, including natural, structured, and specialized categories such as medical or satellite imagery. The tasks span various objectives, comprising object and scene recognition, distance classification, and counting. Consequently, VTAB-1k emerges as a highly valuable resource catering to the needs of both discriminative and generative transfer learning tasks.

In Table 5, we provide information on 19 tasks of the VTAB dataset, including the number of classes and the number of images in each data split of VTAB. Images in the VTAB benchmark encompass three distinct domains: (1) Natural images captured using standard cameras, (2) Specialized images captured using non-standard cameras like those in remote sensing and medical applications, and (3) Structured images generated through simulation environments. VTAB-1k is a subset of VTAB. It contains only 1000 training and validation samples, which are designed for few-shot transfer learning.

**Dreambooth Dataset.**  The DreamBooth dataset Ruiz et al. (2023) focuses on fine-tuning large pre-trained text-to-image diffusion models for personalized subject-driven image generation. This dataset supports the development and evaluation of methods, which enable the generation of novel, photorealistic images of a specific subject in diverse contexts based on a few reference images.

**Discriminative and Generative Fine-tuning Tasks.**  As there are emergent needs to customize the pre-trained foundation models for downstream tasks, a large corpus of fine-tuning methods have been proposed for both both discriminative and generative tasks. Among them, Ruiz et al. (2023); Miao et al. (2024a;b); Xiong et al. (2024); Tarrés et al. (2024); Song et al. (2024b;a) have focused on tuning pre-trained image diffusion models for personalized generation, diversity, compositional generation, and human preference. While Hu et al. (2021); Chen et al. (2022); Karimi Mahabadi et al. (2021) propose to tune propose to tune vision transformers for downstream discriminative tasks.

**Parameter Efficient Fine-tuning.**  To efficiently adapt large models with limited resources. LoRA (Hu et al., 2021) decomposes the weight matrix along the channel dimension into two low-rank matrices, which induce much lower costs in fine-tuning compared with tuning the original full-rank weight. Our filter composition method takes inspiration from (Qiu et al., 2018b), where the multidimensional weights have been decomposed between the spatial and channel dimensions, which has shown effectiveness in continual learning (Miao et al., 2021a; Chen et al., 2023; Liu et al., 2025; Liu et al.), video representation learning (Miao et al., 2021b), graph learning (Cheng et al., 2021a), and generative tasks (Wang et al., 2021a;b). There are some other works on fine-tuning the SVD decomposition of weights(Han et al., 2023), Kronecker decomposition (Patel et al., 2024), sparsity (Wang et al., 2024), non-linearity (Zhong et al., 2024), or fine-tunig bias parameters(Xie et al., 2023).

## A.2  EXPERIMENTAL SETTINGS

**Baseline Methods.**  We compare our method to 7 PEFT approaches: (i) LoRA (Hu et al., 2021), involving the introduction of a low-rank structure of accumulated gradient update by decomposing it as up-projection and down-projection [4]. (ii) LoHa (YEH et al., 2023) utilizes the Hadamard product across two sets of low-rank decompositions to elevate the rank of the resultant matrix and reduce the approximation error. (iii) LoKr (YEH et al., 2023) introduces the Kronecker product for matrix decomposition to reduce the tunable parameters [5]. (iv) BitFit (Zaken et al., 2022) fine-tunes the bias term of each layer. (v) DiffFit (Xie et al., 2023) fine-tunes the bias term, as well as the layer norm

---

[4]LoRA implementation: `https://github.com/microsoft/LoRA`
[5]LoHa and LoKr implementation: `https://github.com/KohakuBlueleaf/LyCORIS`

Table 5: Information of VTAB dataset.

| Dataset | classes | train | val | test | all |
|---|---|---|---|---|---|
| Caltech-101 | 102 | 2754 | 306 | 6084 | 9144 |
| CIFAR-100 | 100 | 45000 | 5000 | 10000 | 60000 |
| Clevr (object distance) | 6 | 63000 | 7000 | 15000 | 85000 |
| Clevr (count) | 8 | 63000 | 7000 | 15000 | 85000 |
| Diabetic Retinopathy | 5 | 35126 | 10906 | 42670 | 88702 |
| DMLab | 6 | 65550 | 22628 | 22735 | 110913 |
| Dsprites (x position) | 16 | 589824 | 73728 | 73728 | 737280 |
| Dsprites (orientation) | 16 | 589824 | 73728 | 73728 | 737280 |
| DTD | 47 | 1880 | 1880 | 1880 | 5640 |
| EuroSAT | 10 | 16200 | 5400 | 5400 | 27000 |
| Flowers102 | 102 | 1020 | 1020 | 6149 | 8189 |
| Kitti | 4 | 6347 | 423 | 711 | 7481 |
| Patch Camelyon | 2 | 262144 | 32768 | 32768 | 327680 |
| Pet | 37 | 2944 | 736 | 3669 | 7349 |
| Resisc45 | 45 | 18900 | 6300 | 6300 | 31500 |
| Smallnorb (azimuth) | 18 | 24300 | 12150 | 12150 | 48600 |
| Smallnorb (elevation) | 9 | 24300 | 12150 | 12150 | 48600 |
| SUN397 | 397 | 76128 | 10875 | 21750 | 108753 |
| SVHN | 10 | 65931 | 7326 | 26032 | 99289 |

and the scale factor of each layer [6]. (vi) OFT and COFT (Qiu et al., 2023) adapts the diagonal blocks of weight matrices to achieve orthogonal fine-tuning [7].

### A.2.1 GENERATIVE TASKS

**Stable diffusion checkpoints.** The pre-trained checkpoint we choose for Stable Diffusion is stable-diffusion-v1-4, which can be found at `https://huggingface.co/CompVis/stable-diffusion`.

**Text prompts for the few-shot generative task.** We adapt the text prompts from YEH et al. (2023) to generate images for Figure 5. Additionally, we use text prompts from Dreambooth (Ruiz et al., 2023) to generate the images and get evaluation results in Table 2.

**Text prompts for the full generative task.** We use specific text prompts to train the Stable Diffusion or generate the images. We list the example prompts for each dataset as follows:

- Caltech-101: This is a picture of accordion.
- CIFAR-100: This is a picture of apple.
- Clevr: This is a picture from CLEVR dataset.
- Diabetic Retinopathy: This is a retina image with no diabetic retinopathy.
- DMLab: This is a picture from DMLab dataset.
- Dsprites: This is a picture from dSprites dataset.
- DTD: This is a picture of banded texture.
- EuroSAT: This is a satellite picture of annual crop.
- Flowers102: This is a picture of pink primrose.
- Kitti: This is a picture from KITTI dataset.
- Patch Camelyon: This is a histopathologic scans without tumor.
- Pet: This is a picture of Abyssinian cat.
- Resisc45: This is a remote sensing picture of airplane.

---

[6]DiffFit and BitFit implementation: `https://github.com/mkshing/DiffFit-pytorch`
[7]OFT and COFT implementation from PEFT library `https://github.com/huggingface/peft`

Table 6: Number of parameters of different PEFT methods.

| | Conv. | Param. | Attn. | Param. |
|---|---|---|---|---|
| Original | $c'ckk$ | $3,686,400$ | $4c^2$ | $1,638,400$ |
| LoRA | $c'kr + ckr$ | $30,720$ | $8cr$ | $40,960$ |
| LoHa | $2c'kr + 2ckr$ | $61,440$ | $16cr$ | $81,920$ |
| Lokr | $c'k + ck + r^2$ | $3,904$ | $8c + 4r^2$ | $5,378$ |
| OFT | $c'ckk/r$ | $460,800$ | $4c^2/r + 4c$ | $207,360$ |
| Ours ($\mathbf{D}$ or $\mathbf{D}_c$) | $mk^2$ | $81$ | $4mk_c^2$ | $576$ |
| Ours ($+\boldsymbol{\beta}$) | $mm_1k^2 + c'mm_1$ | $17,523$ | $4mk_c^2$ | $576$ |

- Smallnorb: This is a picture from SmallNORB dataset.

- SUN397: This is a picture of abbey.

- SVHN: This is a picture of street view house number 0.

## B  ADDITIONAL ANALYSIS

**Computational Time.** The decomposition process using the ISTA algorithm for convolutional atoms and atom coefficients takes about 1 second for each layer and 20 seconds for the whole model, with the code implemented on a GPU. This time is negligible compared to the training duration, which is approximately 60 minutes.

Additionally, we only need to perform sparse coding once for each pre-trained model. The decomposed coefficients can then be reused across all fine-tuning tasks, further reducing the computational cost.

**Computational Cost.** We estimate the computation cost in terms of FLOPs for solving the sparse coding problem: $\min \frac{1}{2}||\mathbf{W} - \boldsymbol{\alpha}\mathbf{D}||_2^2 + \lambda||\boldsymbol{\alpha}||_1$, where we aim to obtain atom coefficients $\boldsymbol{\alpha}$ and atoms $\mathbf{D}$ from the pre-trained weights $\mathbf{W}$. Here $\boldsymbol{\alpha} \in \mathbb{R}^{c'c/k^2 \times m}$, $\mathbf{D} \in \mathbb{R}^{m \times k^2}$, $\mathbf{W} \in \mathbb{R}^{c' \times c}$, $c'$ and $c$ are the numbers of input and output channels, $k$ is the kernel size, $m$ is the number of filter atoms. Suppose ISTA requires $K$ iterations, the FLOPs required for this algorithm is $K(4c'cm + c'c + 6mk^2)$.

In comparison, given the input data $\mathbf{X} \in \mathbb{R}^{B \times c'}$ with batch size $B$, the FLOPs required for one linear layer $\mathbf{Z} = \mathbf{W}\mathbf{X} + \mathbf{b}$, where $\mathbf{W} \in \mathbb{R}^{c' \times c}$ is $6Bc'c + 4Bc + c'c + c$ which includes $2Bc'c + 2Bc$ (forward pass), $4Bc'c + Bbc$ (backward pass) and $c'c + c$ (update parameters).

Suppose we have $c' = c = 512$, $k = 4$, $B = 64$, $m = 9$, with one iteration the computational cost of the decomposition is approximately 9.7 MFLOPs, while the computational cost of one linear layer is 101 MFLOPs.

**Number of Parameters.** We estimate the parameter numbers of different PEFT methods by considering two types of layers as examples: convolutional layers with dimensions $(c', c, k, k)$, and attention layers with parameters $\mathbf{W}_q$, $\mathbf{W}_k$, $\mathbf{W}_v$, $\mathbf{W}_o$, which have dimensions $(c, c)$. Table 6 lists the PEFT fine-tuning methods along with their corresponding parameter counts. Suppose $c' = c = 640$, $k = 3$, the hyper-parameter for other approach is $r = 8$, the hyper-parameters for our method are $k_c = 4, m = 9, m_1 = 3$.

In Table 6, "Ours ($\mathbf{D}$ or $\mathbf{D}_c$)" refers to our method with tuning filter atoms $\mathbf{D}$ and atoms in the linear layer $\mathbf{D}_c$, while "Ours ($+\boldsymbol{\beta}$)" indicates that, in addition to tuning filter atoms, we also incorporate overcomplete filter atoms and their coefficients $\boldsymbol{\beta}$. Compared to other approaches, our method requires the least number of parameters. To determine the parameter counts reported in the paper, we enumerate all the model parameters and sum those that require gradients.

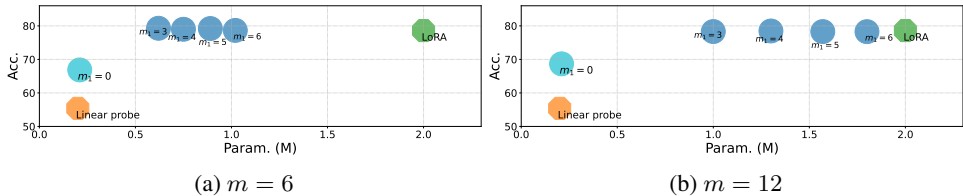

(a) $m = 6$                      (b) $m = 12$

Figure 6: The relations between accuracy and number of fine-tuning parameters, with different numbers of filter atoms ($m = 6$ and $m = 12$).

Table 7: Performance comparisons on the VTAB-1k benchmark with ConvNeXT models pre-trained on ImageNet-21K.

|  | CIFAR-100 | Params. (M) | ImageNet-1k | Params. (M) |
|---|---|---|---|---|
| Full Fine-tuning | **94.1** | 87.7 | **85.8** | 88.9 |
| Linear Probe | 88.6 | 0.1 | 84.7 | 1.0 |
| LoRA (Hu et al., 2021) | 89.2 | 20.1 | 84.8 | 21.0 |
| Ours | _91.8_ | 0.3 | _84.9_ | 1.2 |

## C  ADDITIONAL EXPERIMENTAL RESULTS

### C.1  VALIDATION EXPERIMENTS

We compare different configures of our methods with the following experimental settings and $\alpha$, $\alpha_c$ are always fixed: (1) When fine-tuning $\mathbf{D}$, we experiment with different numbers of filter atoms $m$ in the range of $[6, 9, 12]$. (2) When fine-tuning $\mathbf{D}$ and $\mathbf{D}_c$, we choose $(k'_c, k_c) \in \{(2, 2), (4, 4)\}$, and the number of filter atoms is $m = 9$. (3) When fine-tuning $\mathbf{D}_1$ and $\boldsymbol{\beta}$, we choose $m = 9$, and $m_1$ in the range of $[3, 4, 5]$. (4) When fine-tuning $\mathbf{D}_c$, $\mathbf{D}_1$ and $\boldsymbol{\beta}$, we choose $m = 9$, $m_1 = 3$ and $(k'_c, k_c) = (2, 2)$. We provide additional experiments with $m = 6, 12$ in Figure 6. As we increase $m$ from 6 to 12, the accuracy improves from 66.86% to 68.68%.

### C.2  ADDITIONAL EXPERIMENTS OF DISCRIMINATIVE TASKS

**Full Dataset Fine-tuning.**  For CIFAR-100 and ImageNet-1K, we follow the fine-tuning setting of ConvNeXt in (Lian et al., 2022). We employ the AdamW (Loshchilov & Hutter, 2018) optimizer to fine-tune models for 100 epochs for CIFAR-100, and 30 epochs for ImageNet-1K. The cosine decay strategy is adopted for the learning rate schedule, and the linear warm-up is used in the first 10 epochs for CIFAR-100 and 5 epochs for ImageNet-1K.

We compare the performance of our approach with other baseline methods, and the results on CIFAR-100 and ImageNet-1K are shown in Table 7. With full dataset fine-tuning, the full fine-tuning achieves the highest accuracy, outperforming the parameter-efficient fine-tuning methods. One possible reason is both datasets have sufficient data to prevent over-fitting of the model. Our method achieves a higher accuracy than LoRA while requiring only a small number of parameters (1.2M v.s. 21M). In contrast, in the VTAB-1k benchmark, the amount of data is not very large (e.g., only 1,000 training images), which might cause over-fitting of the model for the full fine-tuning.

**Few-shot Results of ViT.**  We also present the results of ViT in the Table 8. Compared to SSF (Lian et al., 2022), FacT (Jie & Deng, 2023), and Adapter (Houlsby et al., 2019), our method achieves higher average accuracy while keeping the number of tuned parameters minimal.

### C.3  RESULTS OF FEW-SHOT GENERATIVE TASKS

We provide more experimental results of few-shot generative learning learned on concepts "castle" and "canal" in Table. 9 and 10. In this experiment, we also include LoRA, LoHa, and LoKr with different configurations.

Table 8: Performance comparisons on the VTAB-1k benchmark with ViT-B/16 models pre-trained on ImageNet-21K.

| | Natural | | | | | | | Specialized | | | | Structured | | | | | | | | | |
|---|---|---|---|---|---|---|---|---|---|---|---|---|---|---|---|---|---|---|---|---|---|
| | Caltech101 | CIFAR100 | DTD | Flowers102 | Pets | SVHN | Sun397 | Patch Camelyon | EuroSAT | Resisc45 | Retinopathy | Clevr/count | Clevr/distance | DMLab | KITTI | dSprites/loc | dSprites/ori | SmallNORB/azi | SmallNORB/ele | Mean | Params. (M) |
| Full fine-tuning | 87.7 | 68.9 | 64.3 | 97.2 | 86.9 | 87.4 | 38.8 | 79.7 | 95.7 | 84.2 | 73.9 | 56.3 | 58.6 | 41.7 | 65.5 | 57.5 | 46.7 | 25.7 | 29.1 | 65.57 | 85.84 |
| Linear probing | 85.0 | 63.4 | 63.2 | 97.0 | 86.3 | 36.6 | 51.0 | 78.5 | 87.5 | 68.6 | 74.0 | 34.3 | 30.6 | 33.2 | 55.4 | 12.5 | 20.0 | 9.6 | 19.2 | 52.94 | 0.04 |
| Adapter (Houlsby et al., 2019) | 86.1 | **74.1** | 63.2 | 97.7 | 87.0 | 34.6 | 50.8 | 76.3 | 88.0 | 73.1 | 70.5 | 45.7 | 37.4 | 31.2 | 53.2 | 30.3 | 25.4 | 13.8 | 22.1 | 55.82 | 0.27 |
| FacT (Jie & Deng, 2023) | 90.6 | 70.6 | 70.8 | 99.1 | 90.7 | 88.6 | **54.1** | 84.8 | 86.2 | 84.5 | 75.7 | **82.6** | **68.2** | 49.8 | 80.7 | **80.8** | 47.4 | **33.2** | **43.0** | 73.23 | **0.11** |
| SSF (Lian et al., 2022) | 92.6 | 69.0 | **75.1** | **99.4** | 91.8 | 90.2 | 52.9 | **87.4** | 95.9 | **87.4** | 75.5 | 75.9 | 62.3 | **53.3** | 80.6 | 77.3 | 54.9 | 29.5 | 37.9 | 73.1 | 0.24 |
| Ours | **96.3** | 70.5 | 74.4 | **99.4** | **92.1** | **90.4** | 52.7 | 85.9 | **96.0** | 88.6 | **75.8** | 77.4 | 62.2 | 53.0 | **82.6** | 78.1 | **55.1** | 31.7 | 35.9 | **73.77** | 0.22 |

Table 9: Evaluate different approaches in learning the concept ⟨castle⟩.

| | LoHa | | LoRA | | | LoKr | | DiffFit | BitFit | Ours | | |
|---|---|---|---|---|---|---|---|---|---|---|---|---|
| | $d=16$ | $d=4$ | $r=16$ | $r=4$ | $r=1$ | $f=16$ | $f=4$ | | | $m=6, k_c=4$ | $m=9, k_c=8$ | $m=9, k_c=8, m_1=3$ |
| Fidelity | **0.73** | **0.73** | 0.71 | 0.67 | 0.7 | **0.73** | 0.65 | 0.57 | 0.44 | 0.44 | 0.62 | 0.72 |
| Diversity | 3.51 | 3.51 | 4.85 | 5.39 | 4.96 | 4.27 | 6.98 | 10.38 | 16.8 | **16.82** | 8.97 | 8.53 |
| T2I Alignment | 0.21 | 0.21 | 0.23 | 0.23 | 0.23 | 0.23 | 0.25 | 0.26 | 0.25 | **0.31** | 0.28 | 0.27 |
| Param. (M) | 33.86 | 8.47 | 22.67 | 5.67 | 1.42 | 1.12 | 1.06 | 0.58 | 0.34 | **0.05** | 0.75 | 2.39 |

The generated images of different fine-tuning methods are shown in Figure 7 and 8.

## C.4 VISUALIZATION OF GENERATED IMAGES

We visualize images generated by the models trained on each of VTAB tasks from Figure 9 to Figure 24.

## C.5 GRAD-CAM

To understand the underlying reason for the effectiveness of our approach on convolution-based models, we employ Grad-CAM (Gildenblat & contributors, 2021) on the first block of ResNet50, which are fine-tuned on the CUB dataset (Wah et al., 2011) using the same experimental setting as above. For our method, we compare the experiment setting with $m=9$, which means 9 filter atoms $\Delta \mathbf{D}$ and the setting with $(m, m_1) = (9, 4)$, which means 36 $\Delta \mathbf{D}_1$.

Based on the Grad-CAM visualization in Figure 25, our method exhibits larger active regions compared with LoRA. This observation indicates that our approach benefits from preserving the spatial structure of convolutional layers. When utilizing $\Delta \mathbf{D}_1$, which expands the number of filter atoms, we observe more active regions in the Grad-CAM heatmap. This suggests that the introduction of extra filter atoms potentially captures a wider range of feature maps.

We provide more heatmap visualizations of Grad-CAM from the first block of ResNet50 in Figure 26.

Table 10: Evaluate different approaches in learning the concept ⟨canal⟩.

| | LoHa | | LoRA | | | LoKr | | DiffFit | BitFit | Ours | | |
|---|---|---|---|---|---|---|---|---|---|---|---|---|
| | $d=16$ | $d=4$ | $r=16$ | $r=4$ | $r=1$ | $f=16$ | $f=4$ | | | $m=6, k_c=4$ | $m=9, k_c=8$ | $m=9, k_c=8, m_1=3$ |
| Fidelity | **0.52** | 0.47 | 0.39 | 0.38 | 0.37 | 0.36 | 0.38 | 0.31 | 0.33 | 0.16 | 0.29 | 0.39 |
| Diversity | 6.29 | 12.49 | 15.03 | 15.71 | 16.18 | 18.47 | 19.53 | 26.48 | 21.11 | **38.63** | 24.72 | 24.92 |
| T2I Alignment | 0.15 | 0.18 | 0.19 | 0.20 | 0.20 | 0.22 | 0.21 | 0.24 | 0.23 | **0.29** | 0.25 | 0.26 |
| Param. (M) | 33.86 | 8.47 | 22.67 | 5.67 | 1.42 | 1.12 | 1.06 | 0.58 | 0.34 | **0.05** | 0.75 | 2.39 |

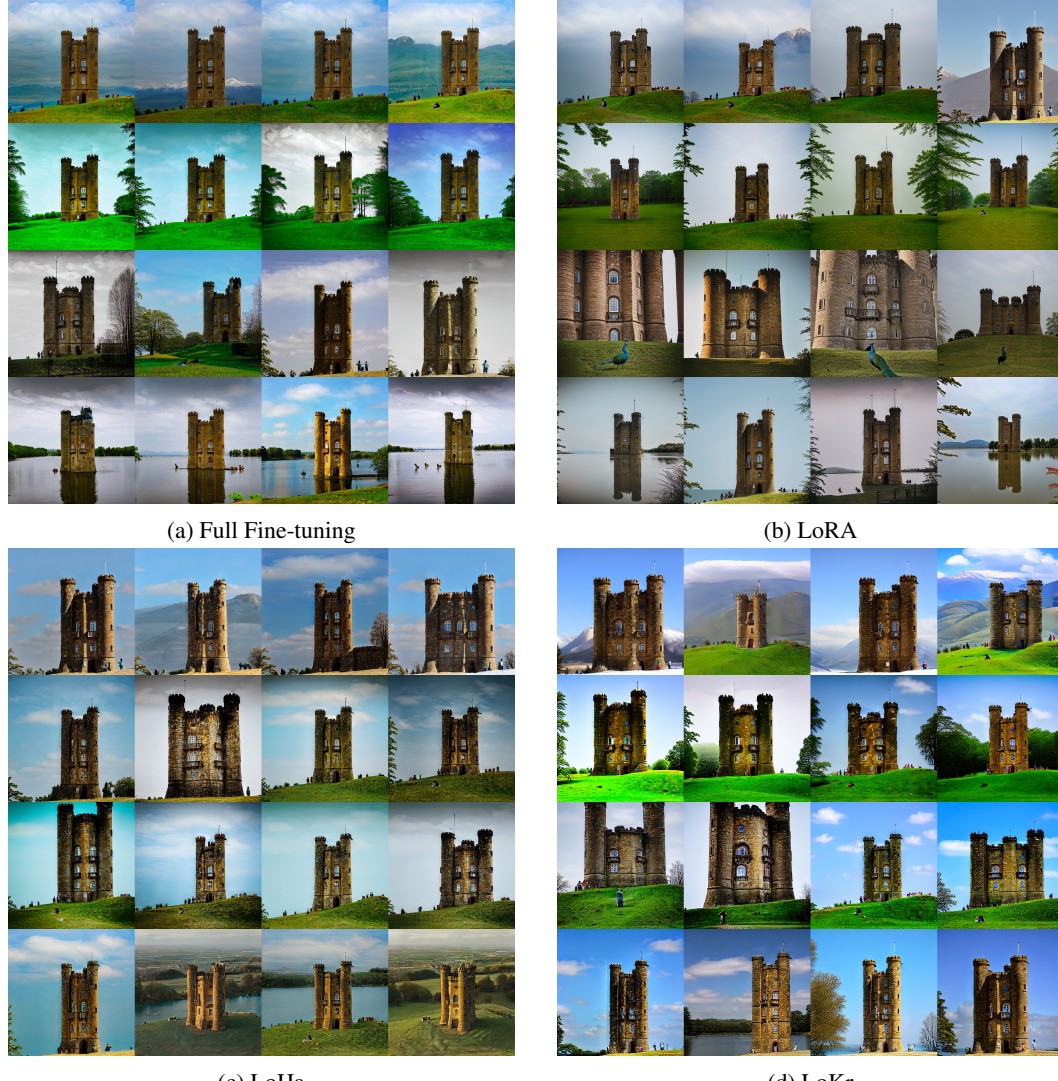

(a) Full Fine-tuning

(b) LoRA

(c) LoHa

(d) LoKr

Figure 7: Images sampled from Stable Diffusion (Rombach et al., 2022) checkpoints fine-tuned with different approaches. The text prompts used to generate images from top to bottom are: "The ⟨castle⟩ stands against a backdrop of snow-capped mountains", "A ⟨castle⟩ surrounded by a lush, vibrant forest", "A peacock in front of the ⟨castle⟩", and 'The ⟨castle⟩ overlooks a serene lake, where a family of geese swims".

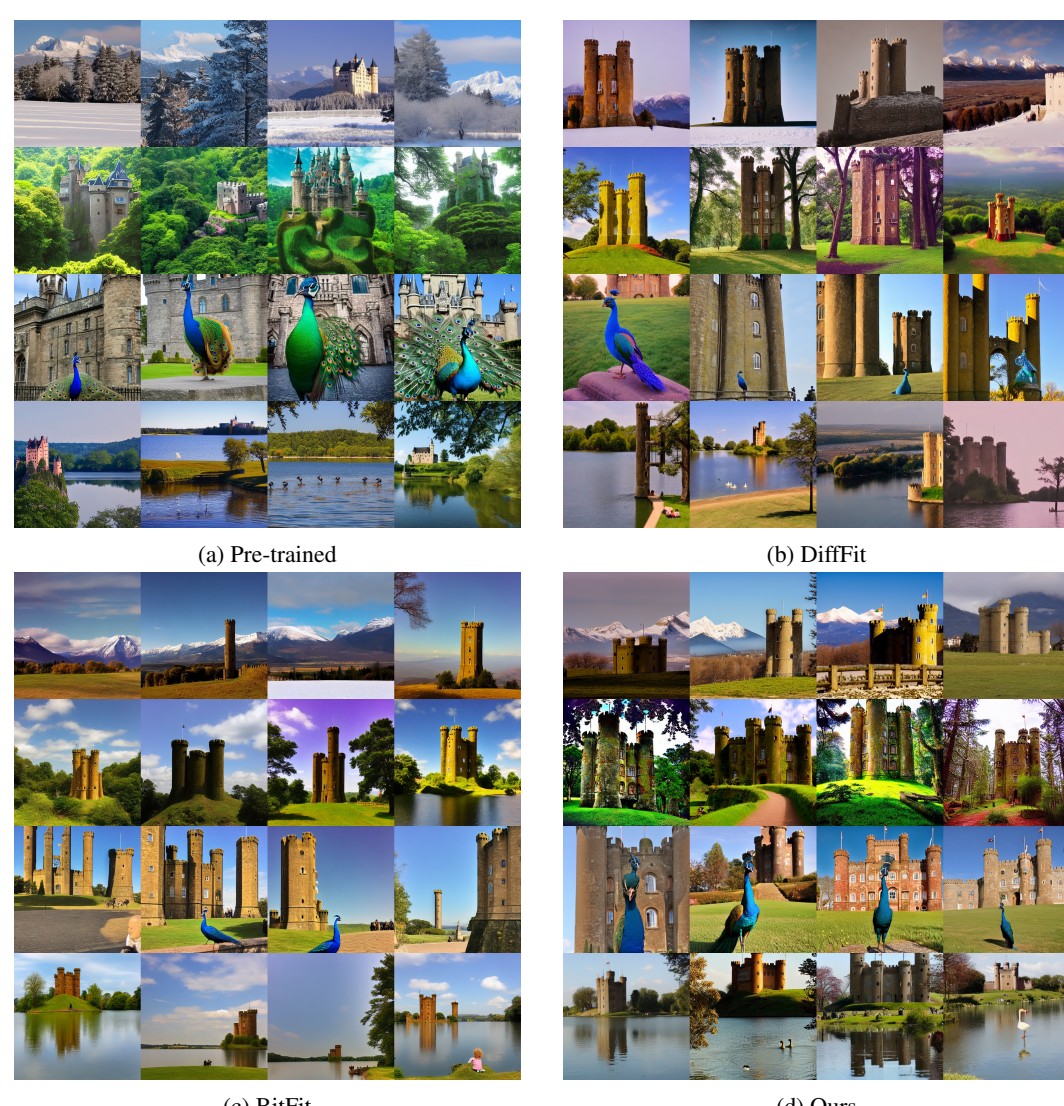

(a) Pre-trained

(b) DiffFit

(c) BitFit

(d) Ours

Figure 8: Images sampled from Stable Diffusion (Rombach et al., 2022) checkpoints fine-tuned with different approaches. The text prompts used to generate images from top to bottom are: "The ⟨castle⟩ stands against a backdrop of snow-capped mountains", "A ⟨castle⟩ surrounded by a lush, vibrant forest", "A peacock in front of the ⟨castle⟩", and 'The ⟨castle⟩ overlooks a serene lake, where a family of geese swims".

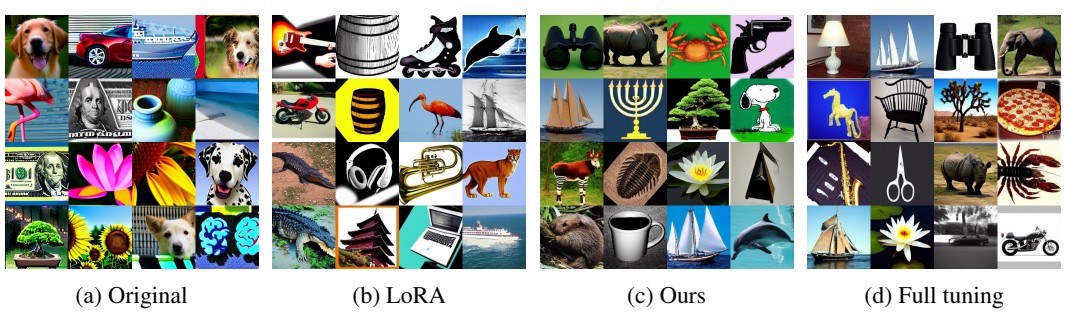

(a) Original      (b) LoRA      (c) Ours      (d) Full tuning

Figure 9: Images sampled from Stable Diffusion checkpoints fine-tuned on the Caltech-101.

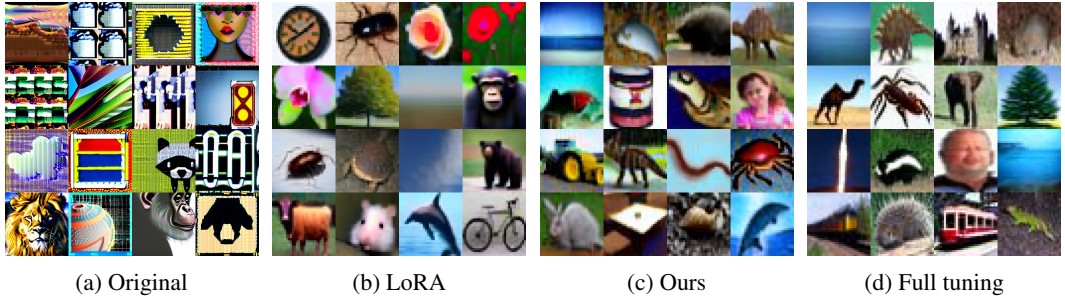

| (a) Original | (b) LoRA | (c) Ours | (d) Full tuning |

Figure 10: Images sampled from Stable Diffusion checkpoints fine-tuned on the CIFAR-100.

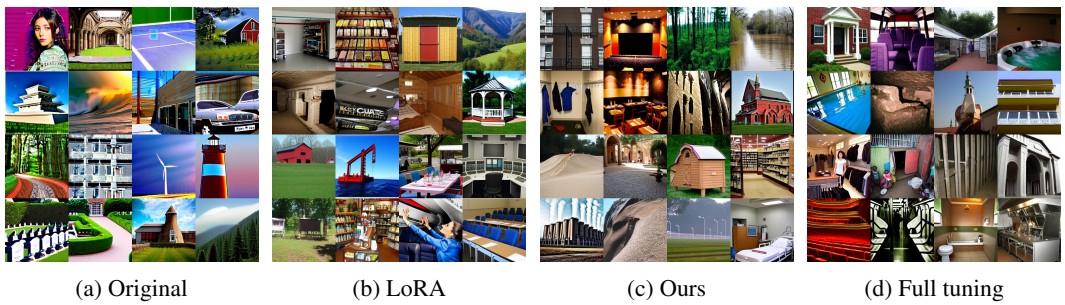

| (a) Original | (b) LoRA | (c) Ours | (d) Full tuning |

Figure 11: Images sampled from Stable Diffusion checkpoints fine-tuned on the SUN397.

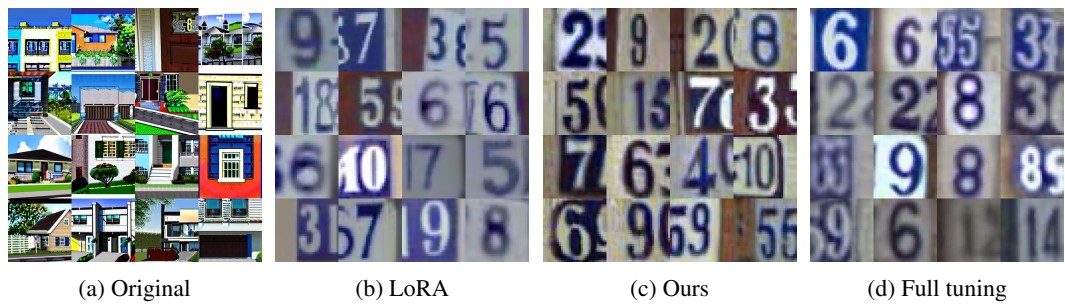

| (a) Original | (b) LoRA | (c) Ours | (d) Full tuning |

Figure 12: Images sampled from Stable Diffusion checkpoints fine-tuned on the SVHN.

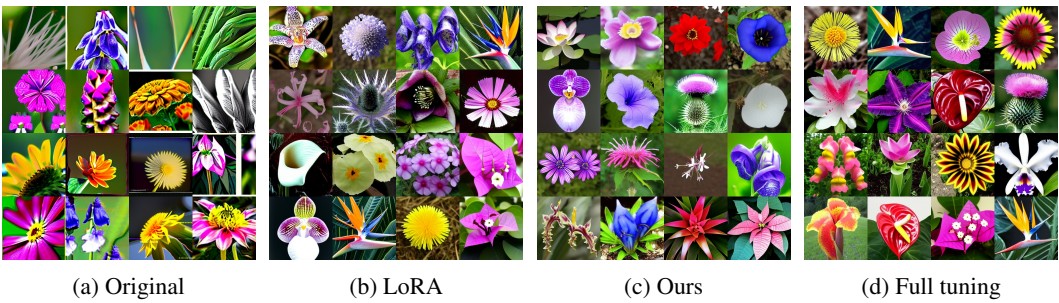

| (a) Original | (b) LoRA | (c) Ours | (d) Full tuning |

Figure 13: Images sampled from Stable Diffusion checkpoints fine-tuned on the Flowers102.

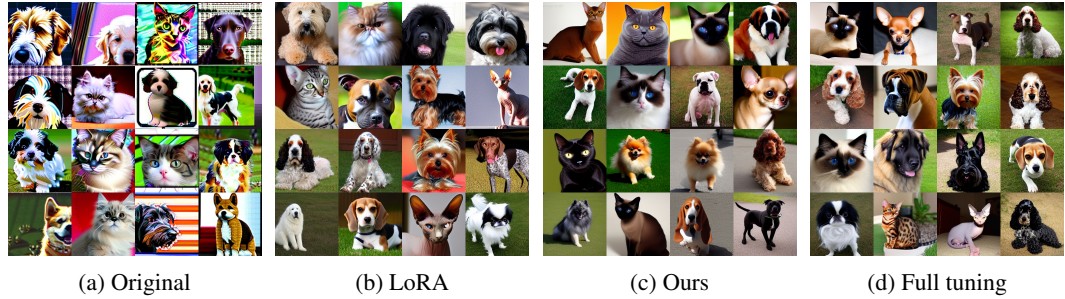

(a) Original        (b) LoRA        (c) Ours        (d) Full tuning

Figure 14: Images sampled from Stable Diffusion checkpoints fine-tuned on the Pets.

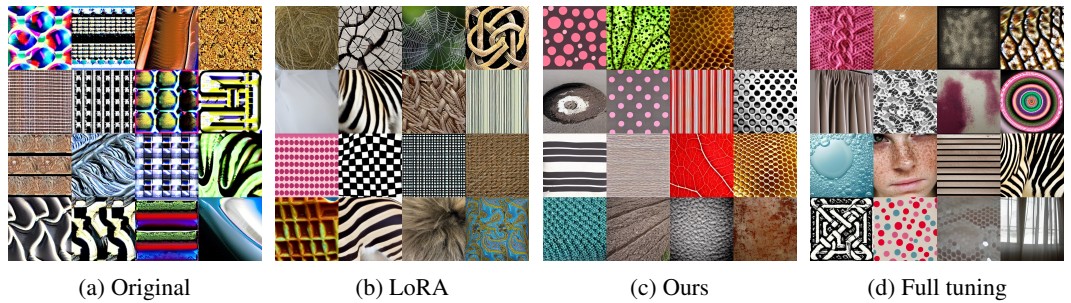

(a) Original        (b) LoRA        (c) Ours        (d) Full tuning

Figure 15: Images sampled from Stable Diffusion checkpoints fine-tuned on the DTD.

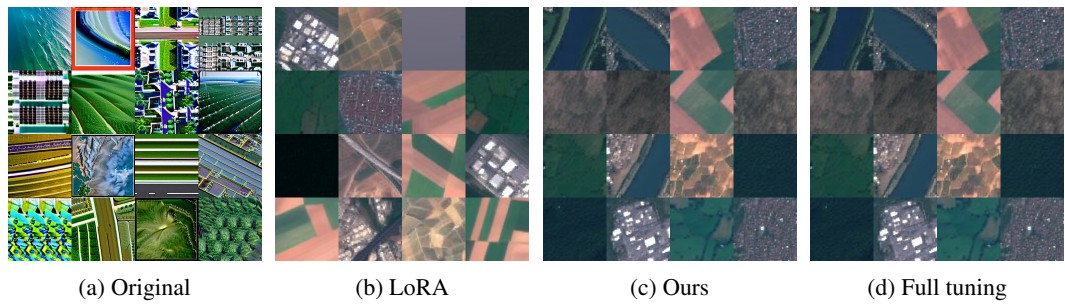

(a) Original        (b) LoRA        (c) Ours        (d) Full tuning

Figure 16: Images sampled from Stable Diffusion checkpoints fine-tuned on the EuroSAT.

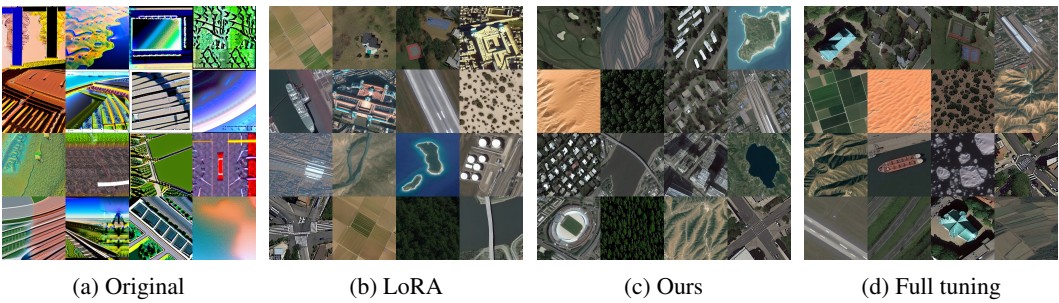

(a) Original        (b) LoRA        (c) Ours        (d) Full tuning

Figure 17: Images sampled from Stable Diffusion checkpoints fine-tuned on the Resisc45.

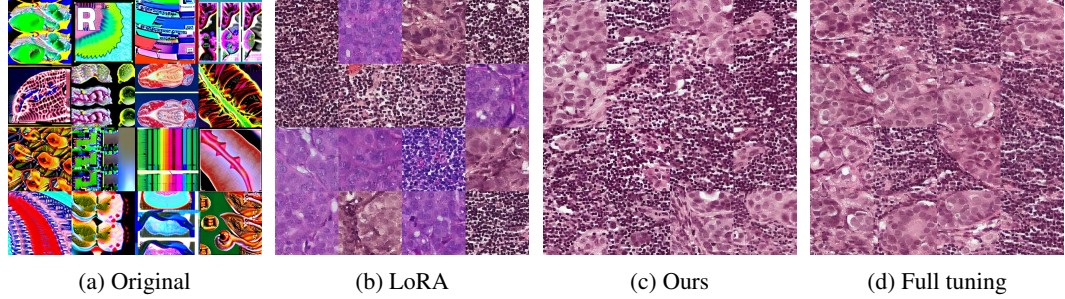

(a) Original      (b) LoRA      (c) Ours      (d) Full tuning

Figure 18: Images sampled from Stable Diffusion checkpoints fine-tuned on the Patch Camelyon.

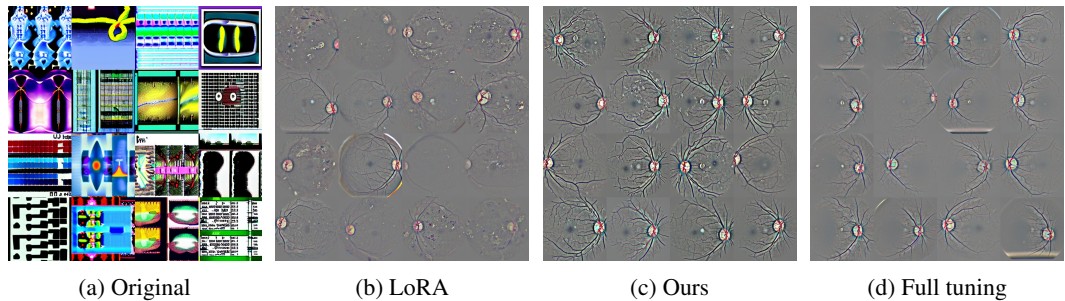

(a) Original      (b) LoRA      (c) Ours      (d) Full tuning

Figure 19: Images sampled from Stable Diffusion checkpoints fine-tuned on the Diabetic Retinopathy.

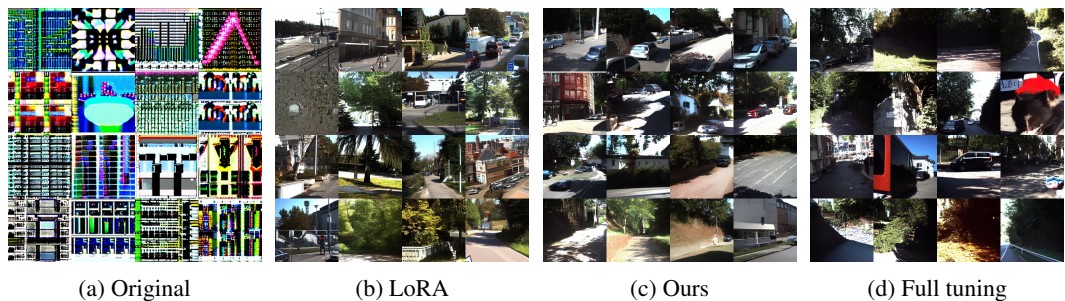

(a) Original      (b) LoRA      (c) Ours      (d) Full tuning

Figure 20: Images sampled from Stable Diffusion checkpoints fine-tuned on the Kitti.

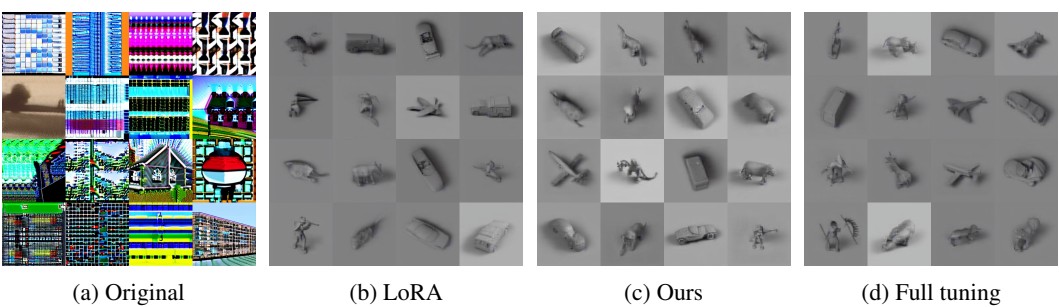

(a) Original      (b) LoRA      (c) Ours      (d) Full tuning

Figure 21: Images sampled from Stable Diffusion checkpoints fine-tuned on the Smallnorb.

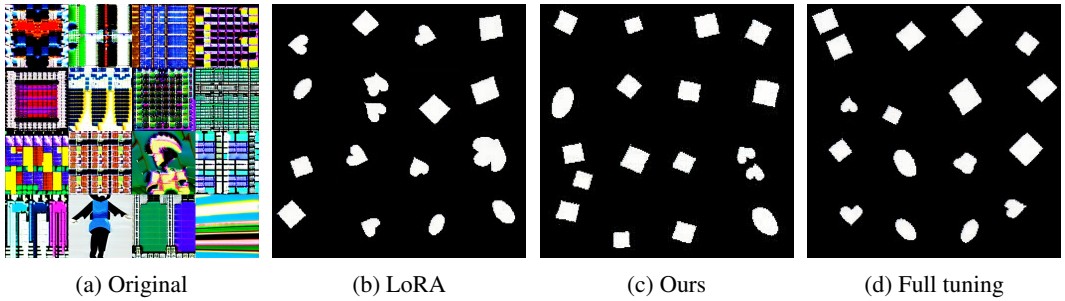

| (a) Original | (b) LoRA | (c) Ours | (d) Full tuning |

Figure 22: Images sampled from Stable Diffusion checkpoints fine-tuned on the Dsprites.

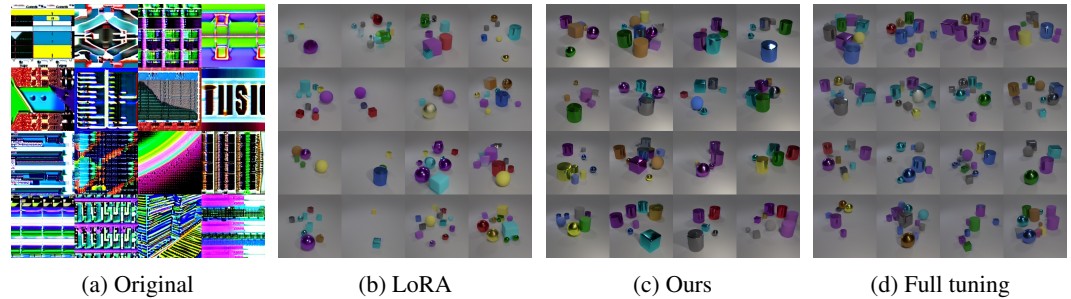

| (a) Original | (b) LoRA | (c) Ours | (d) Full tuning |

Figure 23: Images sampled from Stable Diffusion checkpoints fine-tuned on the CLEVR.

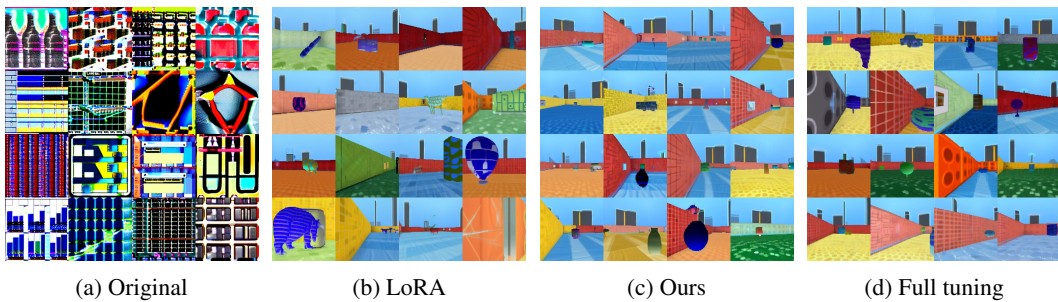

| (a) Original | (b) LoRA | (c) Ours | (d) Full tuning |

Figure 24: Images sampled from Stable Diffusion checkpoints fine-tuned on the DMLab.

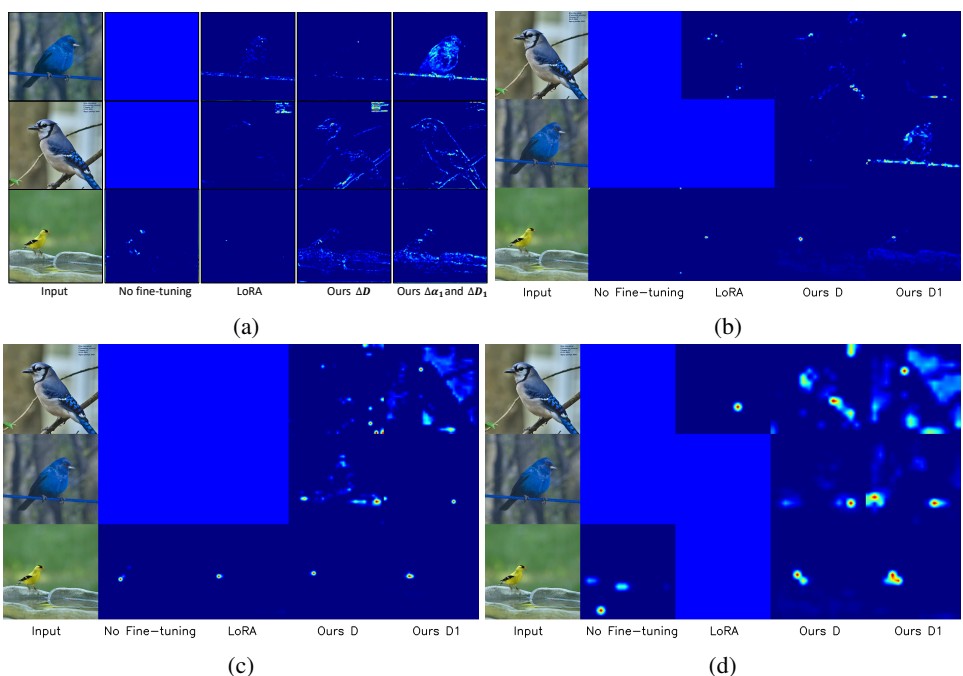

Figure 25: The Grad-CAM heatmap comparisons between our method and LoRA reveal that our approach exhibits larger active regions. The heatmap is generated from the first block of ResNet50 (He et al., 2016) utilizing the CUB dataset (Wah et al., 2011). Fine-tuning the model with $\Delta \mathbf{D}_1$ involves additional filter atoms, which leads to larger active regions in the heatmap compared to fine-tuning $\Delta \mathbf{D}$ only. (a) The Grad-CAM from the first block of ResNet50. (b-d) The Grad-CAM from the 2-4 blocks of ResNet50.

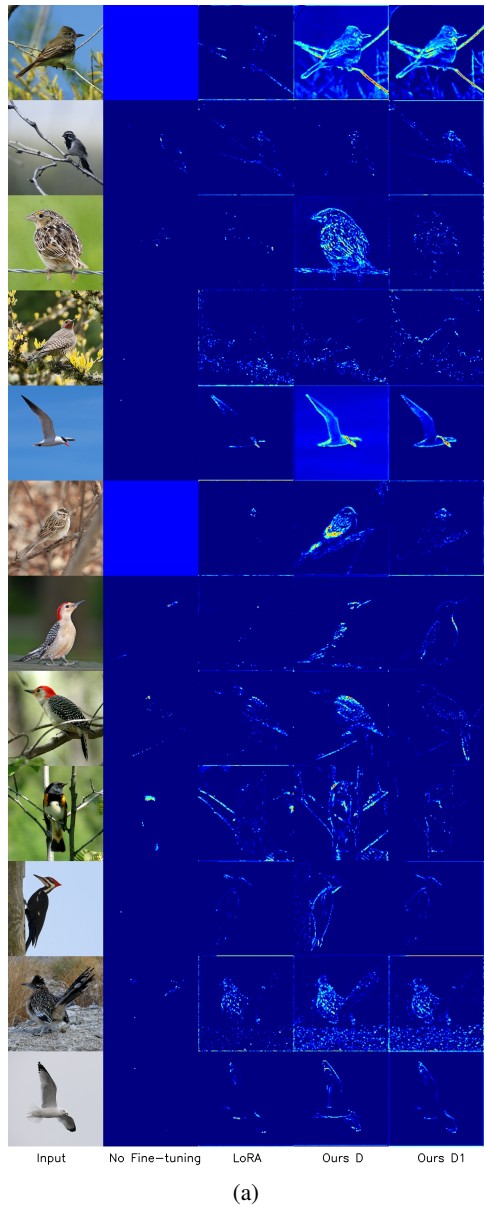

Figure 26: Additional Grad-CAM heatmap comparisons between our method and LoRA from the first block of ResNet50.

