# OpenReview forum: "Large Convolutional Model Tuning via Filter Subspace"
_ICLR.cc/2025/Conference — ICLR 2025 Poster_

### Official Review · Reviewer_gvwM · 2024-10-28

**Soundness:** 2
**Presentation:** 2
**Contribution:** 2
**Rating:** 6
**Confidence:** 5

**Summary:**

This work proposes a PEFT technique for convolution by decomposing the convolutional kernel into spatial and channel components and only fine-tuning the spatial components. Furthermore, the authors introduce a second-order decomposition technique to allow for the training of more parameters. The author validate the effectiveness of this method on various backbone models, such as ResNet50, ConvNeXt, and Stable Diffusion.

**Strengths:**

- The idea of decomposing the convolutional kernel and only fine-tuning the spatial convolution part is interesting, providing options for fine-tuning convolution layers.

- The explanation of the methods and the mathematical formulas are clear.

**Weaknesses:**

- The paper requires additional sparse coding at the start of training to decompose convolutional atoms and coefficient atoms. Due to the need to solve optimization problems, I express concern about its efficiency. The computational cost and time delay associated with this part need to be provided.

- The benchmarks compared in Tables 1 and Table 4 are not up-to-date. LoRA was proposed in 2021, but it is now 2024. To my knowledge, a series of related tasks have been continuously proposed in discriminative tasks in recnet years, such as SSF, FacT, Adapter, Compactor, BinaryAdapter, etc. The authors are encouraged to include the latest methods to demostrate the effectiveness of the proposed method.

- The evaluation metrics for the generation task seem non-standard. It appears that the authors only compared results under one theme image, i.e., the castle. As far as I know, exsting common experimental setups for evaluating subject-driven generation tasks 750 prompt-image pairs, such as in OFT. The experimental setup in this paper only take one subject image, makeing it difficult to prove the effectiveness of the method, especially considering the inherent randomness of diffusion. In addition, I also suggest adding OFT and COFT in the compared methods, which are important and widely used baselines in diffusion model fine-tuning, and are included in the HuggingFace's PEFT library.

**Questions:**

- Besides comparing the number of parameters, what is the GPU memory footprint during fine-tuning for the proposed method? Considering that there is already work indicating that PEFT methods are generally not memory-efficient.

- The idea of decomposing the convolutional kernel and only fine-tuning filter atoms is interesting. However, the experiments in this paper on various tasks do not solid to support the effectiveness of the method. It is necessary to further increase the comparison methods and improve experimental settings. Condidering all the factors, I tend to give a rating of below the acceptance level.

---

> ### Author Response · Authors · 2024-11-23
> **Response to Reviewer gvwM (Part 1)**
>
> We sincerely thank Reviewer gvwM for the constructive comments. We have incorporated these suggestions into the revised manuscript and will continue refining our paper.
>
> **Q1: Provide computational cost and time delay associated with decomposing convolutional atoms and atom coefficients.**
>
> A1: **Computational time:** The decomposition process using the ISTA algorithm for atoms and atom coefficients takes about 1 second for each layer and 20 seconds for the whole model, with the code implemented on a GPU. This time is negligible compared to the training duration, which is approximately 60 minutes.
>
> Additionally, we only need to perform sparse coding once for each pre-trained model. The decomposed coefficients can then be reused across all fine-tuning tasks, further reducing the computational cost.
>
> **Computational cost:** We estimate the computation cost in terms of FLOPs for solving the sparse coding problem: $\min \frac{1}{2} ||W - \alpha D||_2^2 + \lambda ||\alpha||_1$, where we aim to obtain atom coefficients $\alpha$ and atoms $D$ from the pre-trained weights $W$. Here $\alpha \in \mathbb{R}^{c'c/k^2 \times m}$, $D \in \mathbb{R}^{m \times k^2}$, $W \in \mathbb{R}^{c' \times c}$, $c'$ and $c$ are the numbers of input and output channels, $k$ is the kernel size, $m$ is the number of filter atoms.
> Suppose ISTA requires $K$ iterations, the FLOPs required for this algorithm are $K(4c'cm+c'c+6mk^2)$.
>
> In comparison, given the input data $x \in \mathbb{R}^{B \times c'}$ with batch size $B$, the FLOPs required for one linear layer $z=Wx+b$, where $W \in \mathbb{R}^{c' \times c}$ is $6Bc'c+4Bc+c'c+c$ which includes $2Bc'c+2Bc$ (forward pass), $4Bc'c+Bbc$ (backward pass) and $c'c+c$ (update parameters).
>
> Suppose we have $c'=c=512$, $k=4$, $B=64$, $m=9$, with one iteration the computational cost of the decomposition is approximately $9.7$ MFLOPs, while the computational cost of one linear layer is $101$ MFLOPs.
>
> **Q2: Provide comparison with recent methods, such as SSF, FacT, Adapter, Compactor, BinaryAdapter, etc.**
>
> A2: Additional results are presented in the table below. Compared to SSF, FacT, and Adapter, our method achieves higher average accuracy while keeping the number of tuned parameters minimal.
>
> | Method  | C100 | Cal. | DTD | Fl102 | Pets | SVHN | S397 | P.C. | Euro | R45 | Retin. | Cl./c | Cl./d | DM | KITTI | d./loc | d./ori | N/azi | N/ele | params. (M) | avg |
> |---|------|------|-----|-------|------|------|--------|------|----------|--------|-------|-------|-------|------|-------|--------|--------|----------|----------|---------|-----|
> |  Adapter |  **74.1**   |   86.1  |  63.2   |   97.7    |   87.0   |   34.6   |    50.8    |   76.3   |     88.0     |    73.1    |   70.5    |   45.7    |    37.4   |   31.2   |   53.2    |   30.3    |    25.4    |     13.8     |      22.1    |    0.27     |   55.82  |
> |  FacT |  70.6   |   90.6   |  70.8   |   99.1    |   90.7   |   88.6   |    **54.1**    |   84.8   |     **96.2**     |    84.5    |   75.7    |   **82.6**    |    **68.2**   |   49.8   |   80.7    |    **80.8**    |    47.4    |     **33.2**     |      **43.0**    |    0.26     |   73.23  |
> |  SSF |  69.0   |   92.6  |  **75.1**   |   **99.4**    |   91.8   |   90.2   |    52.9    |   **87.4**   |     95.9     |    87.4    |   75.5    |   75.9    |    62.3   |   **53.3**   |   80.6    |    77.3    |    54.9    |     29.5     |      37.9    |    0.24     |   73.10  |
> | Ours  |   70.5   |   **96.3**   |  74.4   |    **99.4**   |    **92.1**  |   **90.4**   |    52.7    |   85.9   |      96.    |    **88.6**    |    **75.8**   |    77.4   |    62.2   |   53.   |   **82.6**    |     78.1   |    **55.1**    |    31.7     |    39.5      |     **0.22**    |  **73.77**  |

---

> > ### Author Response · Authors · 2024-11-23
> > **Response to Reviewer gvwM (Part 2)**
> >
> > **Q3: Provide additional comparison with OFT and COFT on subject-driven generation tasks with 750 prompt-image pairs.**
> >
> >
> > A3: We adopt the experimental setup of OFT and Dreambooth, evaluating our method and the baseline on 30 concepts from Dreambooth. Images are generated using 25 text prompts, resulting in a total of 750 prompt-image pairs. The results are presented in the table below.
> >
> >
> > |               | LoRA  | LoHa  | LoKr  | DiffFit  | BitFit | OFT | COFT | Ours (C1)| Ours (C2)| Ours (C3)|
> > |---------------|-------|-------|-------|----------|--------|-----|------|------|------|------|
> > | Fidelity      |    *0.697*   |    0.693   |   0.693    |     0.622     |     0.571   |   0.656  |   0.652   |   0.594   |   0.652   | **0.707**|
> > | Diversity     |   4.84    |    3.96   |    5.14   |      7.22    |     *10.08*   |   5.86  |   5.92   |   **20.42**   |   9.37   | 6.92 |
> > | T2I Alignment |   0.232    |   0.216    |   0.238    |    0.268      |    0.277    |  0.267   |   0.264   |    **0.301**  |   *0.279*   | 0.236|
> > | Params. (M)      |   22.67    |    8.47   |   1.06    |     0.58     |    *0.34*    |  11.75   |   11.75   |   **0.05**   |   0.75   | 2.39|
> >
> >
> > Compared to other methods, our approach achieves the highest diversity and T2I alignment while requiring a minimal number of tuning parameters with the C1 configuration. Using the C3 configuration, our method attains the highest fidelity among all methods. Additionally, the C2 configuration achieves the second-best T2I alignment while maintaining strong concept fidelity.
> >
> > Compared to OFT and COFT, our method (C2) achieves better T2I alignment (0.279 vs 0.267) and diversity (9.31 vs 5.86) while maintaining similar fidelity (0.652 vs 0.656). Additionally, our method requires significantly fewer tuning parameters (0.75 vs 11.75), as the number of parameters in the atoms is much smaller compared to the rest of the model. In our experiments, the rank of OFT is set to 8, which is the default setting for PEFT.
> >
> > **Q4: What is the GPU memory footprint during fine-tuning for the proposed method?**
> >
> > A4: We have provided the GPU memory requirements of various generative methods in the following table, with a training batch size of 1. Our method requires less GPU memory than most other approaches, primarily due to fine-tuning fewer parameters. With the same training batch size, the intermediate features are similar across methods, but fewer parameters lead to reduced GPU memory usage for storing backward gradients.
> >
> > |               | LoRA  | LoHa  | LoKr  | DiffFit  | BitFit | OFT | COFT | Ours (C2) |
> > |---------------|-------|-------|-------|----------|--------|-----|------|------|
> > | Mem. (MB)      |   8181    |  8027     |   7931    |     7359     |   5433   |  7601   |  7601  |   7333  |
> >
> >
> > **Q5: It is necessary to further increase the comparison methods and improve experimental settings.**
> >
> > A5: We have presented comparisons with Adapter, SSF, and FacT in Q2, as well as with OFT and COFT on 750 prompt-image pairs in Q3. Our method has shown effectiveness compared to baseline approaches, consistent with the experimental results reported in our paper.

---

> ### Comment · Reviewer_gvwM · 2024-11-25
> **Official Comments by Reviewer gvwM**
>
> Thank you for the detailed responds.
>
> My concerns about the additional cost and GPU overhead have been well. addressed. As for the comparison on VTAB and Dreambooth, I have seen the proposed method achieves the best performance.
>
> However, I also noticed the experimental results are not consistent to the original paper. For instance, in the FacT paper, the reported VATB acc is 75.6 with 0.069M #param. But in the rebuttal, the corresponding results are 73.23 with 0.26M #param. So I wonder whether the experimental setup is different here? I still suggest to use the official implementation of previous works to ensure the sota performance of the proposed methods.
>
> In the Dreambooth task, since the OFT paper use the DINO, CLIP-I, CLIP-T and LPIPS as the metrics, what is the performance of the proposed method on these metrics?

---

> > ### Author Response · Authors · 2024-11-25
> > **Response to Reviewer gvwM**
> >
> > Thanks to the reviewer for the follow-up feedback. We provide responses to your questions below.
> >
> > **Q1: Why are the results in the rebuttal different from the FacT paper?**
> >
> > A1: FacT adopts a different method for calculating the average accuracy. In Table 1 of FacT, they use the average of group-wise average accuracy as the final accuracy. Specifically, they first get the averaged accuracy within each group, and then calculate the overall accuracy as the mean of these three averaged accuracy:
> > $[(70.6+90.6+70.8+99.1+90.7+88.6+54.1)/7+(84.8+96.2+84.5+75.7)/4+(82.6+68.2+49.8+80.7+80.8+47.4+33.2+43.0)/8]/3=(80.64+85.3+60.71)/3=75.6$.
> >
> > Following the same process, our method achieves 76.3, which still outperforms FacT. Specifically,
> > $[(70.5+96.3+74.4+99.4+92.1+90.4+52.7)/7+(85.9+96.+88.6+75.8)/4+(77.4+62.2+53.+82.6+78.1+55.1+31.7+39.5)/8]/3=(82.26+86.58+59.95)/3=76.3$.
> >
> > However, to maintain consistency with our paper in the rebuttal, we chose the standard way by calculating the average accuracy across all tasks, following the setting in SSF.
> > Specifically, for FacT,
> > $(70.6+90.6+70.8+99.1+90.7+88.6+54.1+84.8+96.2+84.5+75.7+82.6+68.2+49.8+80.7+80.8+47.4+33.2+43.0)/19=73.23$.
> >
> > For our method,
> > $(70.5+96.3+74.4+99.4+92.1+90.4+52.7+85.9+96.+88.6+75.8+77.4+62.2+53.+82.6+78.1+55.1+31.7+39.5)/19=73.77$.
> >
> > The FacT paper excludes the parameters of the linear head, resulting in 0.069M parameters. To ensure consistency with our paper and SSF, we include the parameters of the linear head, which amount to 0.04M. The revised table is presented below.
> >
> > | Method  | C100 | Cal. | DTD | Fl102 | Pets | SVHN | S397 | P.C. | Euro | R45 | Retin. | Cl./c | Cl./d | DM | KITTI | d./loc | d./ori | N/azi | N/ele | params. (M) | avg |
> > |---|------|------|-----|-------|------|------|--------|------|----------|--------|-------|-------|-------|------|-------|--------|--------|----------|----------|---------|-----|
> > |  Adapter |  **74.1**   |   86.1  |  63.2   |   97.7    |   87.0   |   34.6   |    50.8    |   76.3   |     88.0     |    73.1    |   70.5    |   45.7    |    37.4   |   31.2   |   53.2    |   30.3    |    25.4    |     13.8     |      22.1    |    0.27     |   55.82  |
> > |  FacT |  70.6   |   90.6   |  70.8   |   99.1    |   90.7   |   88.6   |    **54.1**    |   84.8   |     **96.2**     |    84.5    |   75.7    |   **82.6**    |    **68.2**   |   49.8   |   80.7    |    **80.8**    |    47.4    |     **33.2**     |      **43.0**    |    **0.11**     |   73.23  |
> > |  SSF |  69.0   |   92.6  |  **75.1**   |   **99.4**    |   91.8   |   90.2   |    52.9    |   **87.4**   |     95.9     |    87.4    |   75.5    |   75.9    |    62.3   |   **53.3**   |   80.6    |    77.3    |    54.9    |     29.5     |      37.9    |    0.24     |   73.10  |
> > | Ours  |   70.5   |   **96.3**   |  74.4   |    **99.4**   |    **92.1**  |   **90.4**   |    52.7    |   85.9   |      96.    |    **88.6**    |    **75.8**   |    77.4   |    62.2   |   53.   |   **82.6**    |     78.1   |    **55.1**    |    31.7     |    39.5      |     0.22    |  **73.77**  |
> >
> > **Q2: The OFT paper uses the DINO, CLIP-I, CLIP-T and LPIPS as the metrics, what is the performance of the proposed method on these metrics?**
> >
> > A2: Our experiment follows the setup described in (Yeh, et al. ICLR 2024). We use DINO-v2-large to extract image embeddings and evaluate the *fidelity* of the generated images. We employ OpenCLIP (CLIP-ViT-giant), trained on a larger dataset, to assess *T2I alignment*. In comparison, the OFT paper employs DINO-v1-small for the DINO score, while CLIP-T and CLIP-I are based on CLIP-ViT-large.
> >
> > As suggested by the reviewer, we reuse the metrics from the OFT paper and present the results in the table below.
> > Considering the models used in the OFT paper are outdated, we have kept the metrics in our paper unchanged.
> >
> > |               | LoRA  | LoHa  | LoKr  | DiffFit  | BitFit | OFT | COFT | Ours (C1)| Ours (C2)| Ours (C3)|
> > |---------------|-------|-------|-------|----------|--------|-----|------|------|------|------|
> > | DINO      |    0.68   |    0.674   |   *0.682*    |     0.621     |     0.581   |   0.633  |   0.631   |   0.588   |   0.634   |  **0.686** |
> > | CLIP-I     |   0.800    |    *0.801*   |    0.798   |      0.774    |     0.758   |   0.788  |   0.784   |   0.750   |   0.787   | **0.803** |
> > | CLIP-T |   0.209   |   0.203    |   0.212   |    0.232      |    *0.239*    |  0.236   |   0.234   |    **0.248**  |   *0.239*   | 0.205 |
> > | LPIPS      |   0.735    |    0.710   |   0.730   |     0.781     |    *0.796*    |  0.740   |   0.738   |   **0.837**   |   0.788   | 0.731 |
> >
> >
> > Compared to OFT and COFT, our method (C2) achieves a higher CLIP-T (0.239 vs. 0.236), indicating better T2I alignment, and a higher LPIPS (0.788 vs. 0.740), reflecting greater diversity. Our method also maintains good fidelity, as shown by the DINO (0.634 vs. 0.633) and CLIP-I (0.787 vs. 0.788). Furthermore, our approach requires significantly fewer tuning parameters (0.75M vs. 11.75M).

---

> > > ### Comment · Reviewer_gvwM · 2024-11-26
> > > **Official Comment by Reviewer gvwM**
> > >
> > > Thanks for the author response. With the clarified and additional results, my primary concern has been addressed. I still recommend that the authors further refine their writing. For instance, the qualitative presentation in Dereambooth, such as FIg.1, 5,8,9 ,should not constrained to 'castle' only and is recommended to add other subject Additionally, in Tab.4, the comparison on VATB with recent PEFT methods such as FacT, SSF should also be added to offer a wider benchmark.  I will raise my score to 6.

---

> > > > ### Author Response · Authors · 2024-11-26
> > > > **Response to Reviewer gvwM**
> > > >
> > > > Thank you for the suggestions. We have incorporated the feedback provided and included content from the rebuttal in our revised manuscript.

---

### Official Review · Reviewer_GeoC · 2024-10-30

**Soundness:** 2
**Presentation:** 2
**Contribution:** 2
**Rating:** 6
**Confidence:** 3

**Summary:**

This paper proposes to fine-tune large pre-trained models over the filter subspace by only adjusting filter atoms and keeping atom coefficients unchanged for parameter-efficient fine-tuning. To adapt to more complex tasks, the number of tunable parameters in filter subspace is increased to construct an overcomplete set of filter atoms by recursively decomposing each filter atom over another set of filter atoms. Experiments on multiple CNN network architectures across discriminative and generative tasks show the effectiveness of the proposed method.

**Strengths:**

1.Clear motivation to fine-tune the spatial-only filter atoms for PEFT.
2.An interesting idea is to use the overcomplete filter atoms to improve performance.
3.Comprehensive experiments to evaluate the effectiveness of the proposed method.

**Weaknesses:**

1. Spatial-only convolution and cross-channel mixing are similar to group convolution and point-wise convolution. What is the difference when using group convolution and point-wise convolution as filter atoms and coefficients?

2. The authors mainly consider the parameter usage by only fine-tuning filter atoms. I think memory usage and computation are important for PEFT, which should be discussed in this paper for further evaluating the effectiveness of the proposed method. In addition, how to obtain the total parameters of fine-tuning across different networks should be analyzed to improve the readability
3.There are multiple important hyper-parameters (e.g., $m, m_1, k_c$), which significantly affect the final performance. How to set these hyper-parameters.

**Questions:**

See the weaknesses.

---

> ### Author Response · Authors · 2024-11-23
> **Response to Reviewer GeoC**
>
> We sincerely thank Reviewer GeoC for the constructive feedback. We have incorporated most of these suggestions into the revised manuscript and will continue to refine it to further clarify these points.
>
>
> **Q1: What is the difference when using group convolution and point-wise convolution as filter atoms and coefficients?**
>
> A1: The main difference between our approach and group convolution or point-wise convolution lies in its design for parameter-efficient fine-tuning, enabling our method to reconstruct the parameters of the pre-trained model. For instance, in a convolutional layer with $c'$ input channels and $c$ output channels, our method uses filter atoms and atom coefficients to represent the weight update $\Delta F$ as $\alpha \times \Delta D$. In contrast, group convolution and point-wise convolution are unable to represent such weight updates.
>
> In our paper, we further demonstrate that our formulation can be extended to linear weights, a capability that cannot be achieved by group convolution or point-wise convolution.
>
> **Q2: Dicuss the memory usage and computation of the proposed method. How to obtain the total parameters of fine-tuning across different networks?**
>
> A2: **Memory usage.** The GPU memory requirements for various generative methods are shown in the table below, with a training batch size of 1. Our method uses less GPU memory than most other approaches, mainly because it fine-tunes fewer parameters. While intermediate features are similar across methods for the same batch size, fewer parameters result in reduced GPU memory usage for storing backward gradients.
>
> |               | LoRA  | LoHa  | LoKr  | DiffFit  | BitFit | OFT | COFT | Ours (C2) |
> |---------------|-------|-------|-------|----------|--------|-----|------|------|
> | Mem. (MB)      |   8181    |  8027     |   7931    |     7359     |   5433   |  7601   |  7601  |   7333  |
>
> **Computational cost.** The FLOPs for our method is about $4Bc'c/k_c+4Bk_cc'+4Bc+mk_c^2$, where $B$ is the batch size, $c'$ and $c$ are number of input and output channels, $k_c$ is the size of atoms, $m$ is the number of filter atoms.
> Suppose we have $c'=c=640$, $k_c=4$, $m=9$, $B=1$, our method requires only about $0.4$ MFLOPs.
>
> **Number of parameters.** Let's consider two types of layers as examples: convolutional layers with dimensions $(c', c, k, k)$, and attention layers with parameters $W_q$, $W_k$, $W_v$, $W_o$, which have dimensions $(d,d)$.
> The table below lists the PEFT fine-tuning methods along with their corresponding parameter counts. Suppose $c'=c=d=640$, $k=3$, the hyper-parameter for other approach is $r=8$, the hyper-parameters for our method are $k_c=4,m=9, m_1=3$.
>
> |               | Conv.  | Param.  |Attn.  | Param.  |
> |---------------|-------|-------|-------|-------|
> | Original      |   c'ckk    |   3,686,400    |   4d^2    |   1,638,400    |
> | LoRA      |   c'kr + ckr    |   30,720    |   8dr    |   40,960   |
> | LoHa      |   2c'kr + 2ckr    |   61,440    |   16dr    |   81,920   |
> | Lokr      |   c'k + ck + r^2    |   3,904    |   8d+4r^2    |   5,378   |
> | OFT      |   c'ckk/r    |   460,800    |   4d^2/r+4d    |   207,360   |
> | Ours ($D$ or $D_c$)     |   mk^2    |   81    |   $4mk_c^2$    |   576    |
> | Ours (+$\beta$)     |   $mm_1k^2$ + $c'mm_1$    |   17,523    |   $4mk_c^2$    |   576    |
>
> In the table, "Ours ($D$ or $D_c$)" refers to our method with tuning filter atoms $D$ and atoms in the linear layer $D_c$, while "Ours (+$\beta$)" indicates that, in addition to tuning filter atoms, we also incorporate overcomplete filter atoms and their coefficients $\beta$.
>
>
> Compared to other approaches, our method requires the least number of parameters. To determine the parameter counts reported in the paper, we enumerate all the model parameters and sum those that require gradients.
>
> **Q4. There are multiple important hyper-parameters. How to set these hyper-parameters?**
>
> A4: We have conducted an ablation study on these hyper-parameters in Table 1.
>
> We typically set $m=k^2$ to ensure that the reconstructed convolutional kernels from the filter atoms and atom coefficients are full rank, where $k$ represents the kernel size. For instance, for a convolutional layer of size $(c', c, k, k)$, when $k=3$, we set $m=9$.
>
> In Section 4.2, we observe that increasing the hyperparameters $m_1$ and $k_c$ gradually improves accuracy, but also results in more parameters to tune. In our experiments, we find that $m_1=3$ performs well in most cases. For $k_c$, a value of $4$ works effectively for discriminative tasks, while $k_c=16$ is better suited for generative tasks.

---

> > ### Comment · Reviewer_GeoC · 2024-11-28
> >
> > Thanks for your detailed rebuttal. All my concerns are properly addressed. I will raise my score to 6.

---

### Official Review · Reviewer_MS4b · 2024-11-04

**Soundness:** 3
**Presentation:** 3
**Contribution:** 3
**Rating:** 6
**Confidence:** 4

**Summary:**

This paper presents a new way to decompose convolutional layers and experimented a new way to fine-tune large models with those layers by adjusting a small number of parameters based on the decomposition. In particular, the observation that maintaining fixed atom coefficients leads to better results is showed based on the experimental results. Experimental results were compared with other PEFT methods such as LoRA and LoHa and showed interesting results in the provided examples.

**Strengths:**

1. This paper presents an interesting parameter decomposition method to split parameters in large convolutional models.
2. In some situations as shared in the paper, the proposed method can achieve comparable or better results by fine-tuning an even smaller amount of parameters.

**Weaknesses:**

While the proposed decomposition and fine-tuning method is different, this method adjusts parameters in the big model. Comparatively, LoRA serves as a plug-in, which reduces the chance to hurt the capacity of pre-train models.

**Questions:**

Parameter fine-tuning often involves one large pre-trained model and many small tasks. Multiple LoRA's can be plug-in to one model even though there could be conflicts, to solve that scenario. How could this method achieve that?

---

> ### Author Response · Authors · 2024-11-23
> **Response to Reviewer MS4b**
>
> We sincerely thank Reviewer MS4b for the supportive feedback. We have addressed the clarification in the revised manuscript and will continue to refine our paper.
>
>
> **Q1: While the proposed decomposition and fine-tuning method is different, this method adjusts parameters in the big model. Comparatively, LoRA serves as a plug-in, which reduces the chance to hurt the capacity of pre-train models.**
>
> A1: Our method is also used as a plug-in. As shown in Figure 2, our method maintains the parameters $F$ in the large models fixed. We only tune $\Delta F=\alpha \times \Delta D$, which is composed of a fixed coefficient $\alpha$ and tunable filter atom $\Delta D$.
>
> Furthermore, we demonstrate in our paper that our method more effectively preserves the capacity of pre-trained models. For instance, in Table 2, compared to the pre-trained model that generates the most diverse images with the highest T2I alignment, our method maintains high diversity and T2I alignment. In contrast, LoRA overfits to the fine-tuned concept, resulting in significantly lower diversity and T2I alignment.
>
> **Q2: Parameter fine-tuning often involves one large pre-trained model and many small tasks. Multiple LoRA's can be plug-in to one model even though there could be conflicts, to solve that scenario. How could this method achieve that?**
>
> A2: As our method functions as a plug-in, it allows for separate $\Delta D$ for multiple small tasks. Consequently, our approach can exhibit behavior similar to LoRA in handling multiple subtasks.

---

### Meta-Review · Area_Chair_fEBX · 2024-12-21

**Metareview:**

The paper presents the idea of filter subspace tuning, i.e. to fine-tune convolutional neural networks by only adapting spatial filter atoms while leaving the linear combination across channels frozen. The paper presents a clear motivation for the idea, which allows parameter efficient fine-tuning of large models and can achieve results comparable to full fine-tuning as demonstrated on discriminative and generative tasks. After the rebuttal, all reviewers agree on the benefit of the approach.

**Additional Comments On Reviewer Discussion:**

Initially, the reviewers raises several questions regarding for example the comparison to lora, the difference of spatial-only convolutions to group convolutions, details on the memory usage and computational costs and the number of fine-tuning parameters. The rebuttal answered these questions so that all reviewers provide a final score of 6 (two reviewers have raised their score to 6 after the rebuttal).

---

### Decision · Program_Chairs · 2025-01-22

Accept (Poster)